# Freshwater Routing In Eddy-permitting Simulations Of The Last Deglacial : Impact Of Realistic Freshwater Discharge

Ryan Love[1], Heather J. Andres[1], Alan Condron[2], and Lev Tarasov[1]

[1]Department of Physics and Physical Oceanography, Memorial University of Newfoundland, St. John's, Newfoundland, Canada
[2]Geology & Geophysics, Woods Hole Oceanographic Institution, Woods Hole, Massachusetts, USA

**Correspondence:** Ryan Love (rlove@mun.ca)

**Abstract.** Freshwater, in the form of glacial runoff, is hypothesized to play a critical role in centennial to millennial scale climate variability such as the Younger Dryas and Dansgaard-Oeschger Events, but this relationship is not straightforward. Large-scale glacial runoff events, such as Meltwater Pulse 1A, are not always temporally proximal to subsequent large-scale cooling. As well, the typical design of hosing experiments that support this relationship tends to artificially amplify the climate response. This study explores the impact that limitations in the representation of runoff in conventional hosing simulations has on our understanding of this relationship by examining where coastally released freshwater is transported when it reaches the ocean. We focus particularly on the impact of: 1) the injection of freshwater directly over sites of deep-water formation (DWF) rather than at runoff locations (i.e. Hosing), 2) excessive freshwater injection volumes (often by a factor of 5), and 3) the use of present-day (rather than paleo) ocean gateways.

We track the routing of glaciologically-constrained freshwater volumes from four different inferred injection locations in a suite of eddy-permitting glacial ocean simulations using MITGCM under both open and closed Bering Strait conditions. Restricting freshwater forcing values to realistic ranges results in less spreading of freshwater across the North Atlantic and indicates that the freshwater anomalies over DWF sites depend strongly on the geographical location of meltwater input. In particular, freshwater released into the Gulf of Mexico generates a very weak freshwater signal over DWF regions as a result of entrainment by the turbulent Gulf Stream. In contrast, freshwater released into the Arctic with an open Bering Strait or from the Eurasian Ice sheet is found to generate the largest salinity anomalies over DWF regions in the North Atlantic and GIN Seas respectively. Experiments show that when the Bering Strait is open, the Mackenzie River source exhibits more than twice as much freshening of the North Atlantic deep-water formation regions as when the Bering Strait is closed. Our results illustrate that applying a freshwater 'hosing' directly into the North Atlantic with even "realistic" freshwater amounts still over-estimates the amount of terrestrial runoff reaching DWF regions. Given the simulated salinity anomaly distributions and the lack of reconstructed impact on deepwater formation during the Bølling-Allerød, our results support that the majority of the North American contribution to MWP1A was not routed through the Mackenzie River.

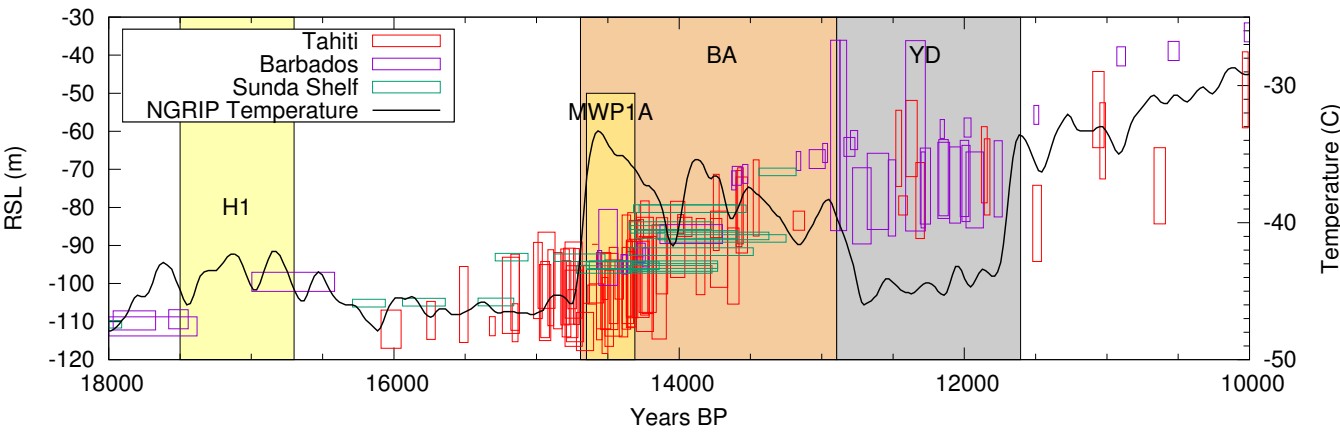

**Figure 1.** Far-field relative sea level records (i.e. sea level relative to present day, where the far-field sea level signal is roughly comparable to eustatic sea level) from Barbados (Fairbanks, 1989), Sunda Shelf (Hanebuth, 2000), and Tahiti (Deschamps et al., 2012) plotted along with a recent NGRIP temperature reconstruction for the deglacial (Kindler et al., 2014).

## 1 Introduction

The most recent deglacial and glacial intervals are punctuated by large-scale centennial to millennial scale climate variability, including the Bølling-Allerød, Younger Dryas, and Dansgaard-Oeschger events. Changes in freshwater discharge into the ocean and subsequent transport are thought to play a significant role in this variability through their resultant impact on deepwater formation (DWF) in the North Atlantic (Broecker et al., 1989; Manabe and Stouffer, 1997; Teller et al., 2002). However, recent earth system modelling (Peltier and Vettoretti, 2014; Zhang et al., 2014; Kleppin et al., 2015; Brown and Galbraith, 2016; Vettoretti and Peltier, 2016; Zhang et al., 2017; Klockmann et al., 2018) has also demonstrated that changing freshwater inputs into the oceans is not required to get such transitions.

Furthermore, there are clear intervals during the last deglaciation when strongly enhanced net freshwater injection into the oceans resulted in no temporally proximal cooling (see Fig. 1). In the case of Meltwater Pulse (MWP) 1-A, current best estimates of its timing indicate that, within dating uncertainties, the freshwater injection aligns with the Bølling-Allerød (Deschamps et al., 2012) warm interval. This is consistent with the physical reasoning that a warm interval coinciding with continental-scale ice sheets should result in enhanced glacial runoff. The onset of the subsequent cold Younger Dryas interval occurs more than a millennium later, which is longer than would be consistent with a direct physical linkage. Understanding the factors that control the impact of freshwater on climate is an important step toward understanding these past climate changes and predicting those in the future.

Climate models have generally supported the ability of freshwater to generate abrupt climate transitions in hosing experiments, where large volumes of freshwater $(1 - 10\text{dSv}^1)$ are imposed over sites of DWF (Kageyama et al., 2013). Such hosings

---

[1]We use the more appropriately scaled unit of dSv, 1/10th of a Sverdrup, to better reflect the magnitude of realistic freshwater fluxes.

are meant to reproduce the effect of changing freshwater input into the oceans from regional ice sheet melt and iceberg discharge as well as rerouting of surface runoff (Tarasov and Peltier, 2005). However, the climate model support for this connection between freshwater injection and climate transitions is problematic given at least three common experimental design problems that likely amplify climate system response compared to that which would ensue from more realistic freshwater injection experiments.

The first issue is the geographic distribution of freshwater injection. Given the transport mechanisms of coastally released freshwater, e.g. boundary currents and mesoscale eddies (Condron and Winsor, 2012; Hill and Condron, 2014; Nurser and Bacon, 2014), are well below the resolution of commonly used models, many studies opt to bypass the transport by injecting freshwater directly onto sites of DWF (e.g. Manabe and Stouffer, 1997; Peltier et al., 2006; Otto-Bliesner and Brady, 2010). Often-times the freshwater is introduced directly over 50-70N or the Ruddiman/Ice-Rafted-Debris (IRD) belt, a region of ocean approximately between 40N and 50N in the North Atlantic, situated between Newfoundland, Canada and Portugal. These most common injection locations usually inhibit DWF through a persistent freshwater cap that results in near-immediate decreases in AMOC (Stouffer et al., 2006). This attempt to compensate for coarse model resolution via hosing is problematic, since it assumes that all of the freshwater reaches the near-surface DWF zone intact. It is unclear if a more realistic representation of runoff routing would yield a similar freshwater signal at the zones of DWF. The only eddy-permitting and boundary-current resolving modelling of freshwater forcing from actual continental outlets to date under glacial boundary conditions suggests this is not the case (Condron and Winsor, 2012; Lohmann et al., 2020). However both of these studies have design limitations which limit the interpretability of their results. The unstructured mesh of FESOM in Lohmann et al. (2020) has refined (but not quite eddy-permitting) grid resolution largely only over the Arctic ocean and at coastal boundaries but is unable to resolve the impact of mesoscale eddies on freshwater transport over the central North Atlantic. In order to offset the short one-year interval of injection, Condron and Winsor (2012) relied on fluxes of freshwater (50dSv) that were more than a factor of 20 larger than estimates derived from glaciological modelling over the Younger Dryas (Tarasov and Peltier, 2005, 2006).

The second significant issue we identify in modelling studies is the use of unrealistic volumes of freshwater in injection experiments. The amount of freshwater which is injected tends to be excessive, often of the order of 10dSv (e.g. Peltier et al., 2006). If one considers the total discharge due to net ice mass loss during MWP1A, then a 10 to 15m rise of sea level over one third of a millennium (Deschamps et al., 2012) translates to a global excess (above background precipitation minus evaporation) discharge of 3 to 5dSv. This is the total contribution from outlets of all ice sheets. For North America, peak centennial-mean values of $1-1.5$dSv for any single discharge sector have been inferred from data-constrained glaciological modelling (Tarasov and Peltier, 2006). For Eurasia, Brendryen et al. (2020) estimates a rate of discharge of 2dSv into the Norwegian Sea and Arctic Ocean from re-assessing Norwegian Sea sediment cores. It is understood that varying scales of freshwater injection can elicit wide ranges in climate behaviour (Roche et al., 2009; Kageyama et al., 2013). A previous investigation covering a range of freshwater injection fluxes shows that the change in North Atlantic Deep Water (NADW) formation becomes less sensitive to the injection location as fluxes grow larger (Roche et al., 2009), due to an increasing amount of diffusive spread. Furthermore, results from Peltier et al. (2006) and Stouffer et al. (2006) demonstrate that the rate of change in AMOC and Greenland surface air temperature is much stronger for a 10dSv injection compared to that of a 3dSv injection.

The third limitation involves the use of present-day rather than paleo-bathymetry, and especially its effect on ocean gateways. The Bering strait in particular provides the sole connection between the Pacific and the Arctic, and has been shown to have global impacts on ocean circulation in previous studies (e.g. Hu et al., 2007, 2012, 2015). As such, we also conducted an additional experiment examining the impact of uncertainties in the state of the Bering Strait. While it is clear that the Bering Strait was closed at the time of MWP1A, there is some evidence that it may have been open during the onset of the Younger Dryas (England and Furze, 2008) although the majority of available evidence indicates closure during this time (e.g. Jakobsson et al., 2017). Hu et al. (2007, 2012, 2015) demonstrate that the transport of freshwater can be strongly affected by the state of the Bering Strait under various background climates, with the effect that a closed Bering Strait leads to a stronger AMOC. Also, when the strait is closed, freshwater injected into the 50-70N band remains in the Arctic Ocean longer and results in a delayed recovery of the AMOC from freshwater forcing. We explore the ambiguity of the Bering Strait for the one key injection region that most likely be affected by its state, the Mackenzie River in the Canadian Arctic.

The goal of this study is to directly address all of these limitations (i.e. unrealistically large volumes of freshwater injection, injection directly onto sites of DWF, and misrepresentation of gateways) via a more realistic experimental design, and to elucidate how these limitations may bias inferences about the connection between glacial runoff and salinity anomalies at sites of deep-water formation. This study provides one of the first assessments to simultaneously address all of these issues, using freshwater injection amounts constrained by the output of a calibrated ensemble of glaciological models (Tarasov and Peltier (2006) and ongoing work) applied to a range of plausible source regions in a suite of simulations that are eddy-permitting over all regions of freshwater transport except the Arctic, where mesoscale eddies tend to have spatial scales of $O(10km)$ or less (Nurser and Bacon, 2014). We achieve this by assessing the amount of freshwater transported to sites of DWF from the main Northern Hemisphere outlets. Given the importance of specific freshwater injection locations, we separately examine freshwater transported from the major outlet regions for the Northern Hemisphere posited to be important for these types of climate transitions: the Mackenzie River (MAK), Fennoscandia (FEN), the Gulf of St. Lawrence (GSL), and the Gulf of Mexico (GOM).

## 2 Experimental Design

We start our discussion of the experimental design with a brief overview of the model configuration, followed by information regarding the simulations conducted, ending with a discussion of the limitations of our configuration. All of the simulations were performed using the Massachusetts Institute of Technology General Circulation Model (MITgcm, Marshall et al., 1997) coupled global sea-ice/ocean model in a Cubed-Sphere 6x510x510 (CS510) configuration, which provides $\approx$ 18km spatial resolution with 50 vertical levels. This grid geometry and resolution is eddy-resolving to eddy-permitting for all ocean regions equator-ward of $60°N$ (Chelton et al., 1998; Nurser and Bacon, 2014). Our configuration is of comparable complexity and resolution to most members of the multi-model ensemble of present-day simulations presented in Hirschi et al. (2020). This resolution means the model is able to capture small-scale phenomena like coastal boundary currents and mesoscale eddies that are among the primary mechanisms responsible for the transport of terrestrial meltwater discharged into coastal, near-shore,

environments (Condron and Winsor, 2012; Hill and Condron, 2014). Most of the coarser resolution models used in current and previous PMIP and CMIP working groups are unable to do this explicitly (Yang, 2003). A sample map of daily-mean salinity is shown in Fig. 2 in order to illustrate the turbulent characteristics of this model configuration. Generally, using eddy-resolving ocean model configurations results in a better representation (primarily greater current velocities) of small-scale features and a better agreement between models and observations for present day (Hirschi et al., 2020). As well, some large-scale features, particularly the subpolar gyre, tend to be stronger at eddy-permitting resolutions (Treguier et al., 2005). An overall increase in the transport speed of tracers at higher resolutions is notable in Weijer et al. (2012). They find that a passive tracer released from the Greenland ice sheet covers the entirety of the subtropical gyre region within 2 years for a configuration at eddying resolution, whereas the non-eddying configuration of the same ocean model takes up to 5 years to even reach the eastern seaboard of North America. These features show that conducting simulations at eddy-permitting resolutions results in an overall more vigorous transport of glacial runoff relative to coarser, non-eddy-permitting resolutions.

All simulations and relevant setups/forcings are listed in the supplemental table S1. The freshwater injection experiments were branched from one of two Younger Dryas control simulations, which were themselves initialized from a Last Glacial Maximum (LGM) simulation. The LGM simulation was run for $\approx 20$ years using MITGCM and the same boundary conditions as Hill and Condron (2014). This initial LGM simulation featured LGM bathymetry, sea level 120m lower than present, a glaciated Barents-Kara Sea and Canadian Archipelago, and a closed Bering Strait. The surface forcing used in the LGM simulation included winds, precipitation, 2m atmospheric temperatures, short and longwave radiation, surface runoff, and humidity from the CCSM3 working group's contribution to PMIP2 (Braconnot et al., 2007). We do not use surface restoration in our experiments. Evaporation is handled internally by the model in the EXF (EXternal Forcing package) based on prescribed precipitation, relative humidity, and surface runoff fields. We used the 3D ocean salinity and temperature fields from the LGM simulation to initialise a control run with Younger Dryas bathymetry (including closed Bering Strait, CBS) and LGM surface forcing, which was integrated forward for an additional 10 years. The open Bering Strait (OBS) Younger Dryas control run was then branched from the CBS run, and both OBS and CBS control runs were integrated forward for an additional 10 years before the freshwater forcing simulations (MAK, FEN, GSL, and GOM) were branched off.

Sea level was adjusted in all Younger Dryas runs to that provided by the sea-level solver component of the Glacial Systems Model of Tarasov et al. (2012) at approx. 13ka. This sea level adjustment is not eustatic as was implemented in Hill and Condron (2014), but included major features which affect the geoid (Mitrovica and Milne, 2003) excepting the rotational component (Milne and Mitrovica, 1998), which has the weakest effect. The largest ensuing ocean gateway change at Younger Dryas compared to LGM is the opening up of the Barents-Kara Seas. Opening both the Barents-Kara Seas and the Bering Strait increases the flow into and out of the glacial Arctic Ocean.

The freshwater injection runs were run for $\approx 20$ years beyond the branch point, during which time the freshwater injection was applied continuously to one of four sites. These sites are the MAK outlet as in Condron and Winsor (2012), the GOM, the GSL, and a region off the coast of Norway in FEN (see table S1 for more details). Figure 2 provides a map showing each injection outlet and the regions over which the salinity averages are calculated. We note that there is overlap between the North Atlantic deep-water formation zone (whose bounds were determined from the mixing region in our model, see Fig. S1) and

the GIN Seas region (whose bounds were obtained from Seidov et al., 2016). As well, the GIN Seas region is more extensive than if it were constrained to what some models exhibit for their regions of deep-water formation in the GIN Seas. For our experiments, 2dSv of freshwater were continually imposed to be an analogue for the outflow of solid and liquid mass from the Northern Hemisphere ice sheets. This is about one and a half to twice the deglacial ice sheet sourced regional discharge flux (both meltwater and iceberg) when considering the centennial-mean peak for any single discharge region (for North America) in the data-constrained glaciological modelling of Tarasov and Peltier (2006), while being consistent with recent estimations for Fennoscandia (Brendryen et al., 2020). While this flux is larger than the centennial-mean peak of Tarasov and Peltier (2006) for North America, this injection rate is both typical of previous studies exploring freshwater forcing during the Younger Dryas (such as Kageyama et al., 2013; Gong et al., 2013) and comparable to the Tarasov and Peltier (2006) estimates for glacial runoff, rather than orders of magnitude greater as previously discussed. Treating the outflow as mostly liquid and neglecting any differences in transport between icebergs and freshwater is a reasonable assumption for MAK, GSL and GOM. MAK and GSL liquid discharge is larger than solid discharge for almost all of the post $\approx 18$ka deglacial interval, especially during discharge peaks (at least for the nn9894 and nn9927 GLAC1-D variants of Tarasov et al., 2012). Similarly, all discharge into GOM would have been liquid during this interval. This assumption is likely not applicable for the flow of mass into the Hudson Strait/Baffin Bay region, where the majority of the discharge was in the form of icebergs until 11ka (at least in the data-constrained glaciological modelling of Tarasov et al., 2012). As such, an investigation exploring the flow of glacial runoff from the Hudson Strait/Baffin Bay region would have to account for the differences in transport of icebergs versus freshwater, so we opted not to test this outlet.

Finally, we draw the reader's attention to three possibly significant experimental design limitations: the short duration of the integrations, issues with the surface forcing, and the uncoupled nature of the ocean simulations. These issues do not negate the value of our experiments for the question of freshwater routing during the deglacial interval for the following reasons. Including the spin-up time, the injection runs are at longest $\approx 30$ years due to computational constraints. By using the same surface forcing in the Younger Dryas simulations as was used in the LGM simulation and initializing from the temperature and salinity fields of the high-resolution LGM simulations, we were able to reduce the necessary spin-up time and thereby make efficient use of computational resources. As a result, these simulations are of sufficient duration to resolve surface ocean dynamical components (Le Corre et al., 2020), particularly the surface transports of freshwater by the glacial ocean, which is our primary focus here. However, they are of insufficient duration to spin up the deeper regions of the ocean, which require millennia. Cold-starting the YD simulations or initializing from present-day would have required substantially longer spin-up before the surface conditions would have been considered "reasonable" approximations to YD.

We expect that the main effects of full equilibration of the deep ocean on the surface transports, relative to our state, would be enhanced overturning in the North Atlantic (see section S3) and a latitudinal shift of the Gulf Stream. Nevertheless, in order to minimize the impact of any residual spin-up issues on the conclusions of this study (see supplemental section S2 for a brief discussion on this topic), the OBS and CBS control simulations were also run forward in time by an amount equal to the injection experiments, so all results are interpreted with respect to changes in the control simulations. Further discussion of

the implications of the short duration of experiments for the AMOC in particular can be found in the supplemental materials section S3.

The Younger Dryas surface forcing fields we use are monthly values derived from a coupled climate model configured for LGM using ICE-5G boundary conditions. This ice sheet configuration has been shown to generate more zonal atmospheric circulation patterns than more recent reconstructions of LGM ice sheets (Ullman et al., 2014), and LGM winds are expected to be stronger and more southward-shifted over the North Atlantic than winds during the Younger Dryas (Andres and Tarasov, 2019; Löfverström and Lora, 2017). These biases are expected to enhance zonal transport in the North Atlantic, as seen in comparison to a sensitivity experiment described in Supplemental Section S5. Thus, we would expect freshwater transported across the North Atlantic to be routed further south than under forcing which does not have this bias.

Lastly, the configuration used is uncoupled and is lacking ocean-atmosphere interactions and feedback processes. At oceanic mesoscales, uncoupled configurations generally exhibit weaker AMOC relative to coupled configurations (Hirschi et al., 2020), and spatially variable sea surface temperature gradients can result in a wind stress curl which itself can modify sea surface temperatures (Chelton and Xie, 2010). However, the expected magnitude of this latter effect is small. For example, our injections can result in sea surface temperature differences of several K (see Fig. S2), which may result in local changes in 10m wind speeds of $\approx \pm 1 \text{m/s}$ (Song et al., 2009).

## 3   Results and Discussion

### 3.1   Control simulation climate

Our two control simulations are very similar, as is expected given they only differ in the state of the Bering Strait. An examination of the salinity fields in Fig. 2 and the velocity fields in Fig. S3 reveals that the Gulf Stream in both simulations is highly zonal. This feature is due to the surface wind forcing as discussed in the Experimental Design section and in Section S5. The sea ice extent is consistent between simulation years, with maxima varying between $15.75 - 16 \text{km}^2$ for the closed Bering Strait (CBS) simulation and $17 - 18 \text{km}^2$ for the open Bering Strait (OBS) simulation. Sea ice extent is larger in the OBS simulation due to two features: the expanded ocean area surrounding the open Bering Strait, and enhanced sea ice export, see Fig. S4. Of note is the large region of sea ice cover off the eastern coast of North America, extending as far south as 40N in the winter (see Fig. 2 and Figs. S3 and S4). The absence of corresponding sea ice cover over the eastern North Atlantic indicates that there is substantial surface heat transport to this region, despite the zonal Gulf Stream. Both the GIN and Labrador Seas are covered with sea ice during the winter. Due to the extensive sea ice in this model, the main DWF zone lies south of the sea ice edge, in the region between Greenland, Iceland, and the British Isles (see also Fig. S1). Mixed-layer depths in the Labrador Sea region are much shallower than in the northern North Atlantic and indicate that not much DWF is occurring there.

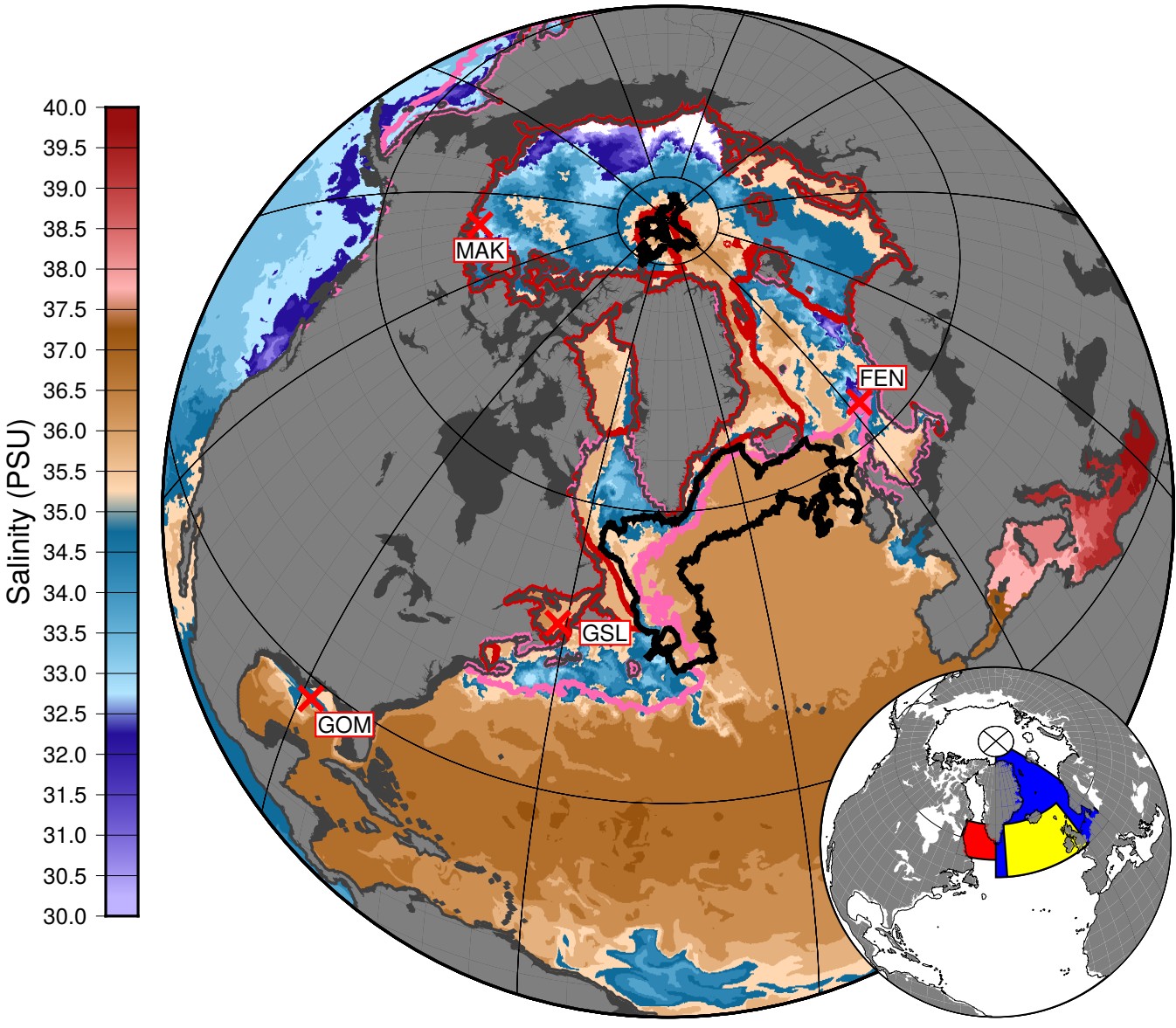

**Figure 2.** Sea surface salinity from a single day (i.e. daily mean data) at the end of the CBS Control run. Present day land-sea mask is shown in light grey while simulation land-sea mask is contoured in a darker grey. The dark red and pink contours denote the minimum and maximum sea ice extent respectively, of at least $15\%$ sea ice coverage calculated over the last 5 years of the simulation. A 1000m mixed layer depth contour from the same time interval as the sea ice extent contours is shown in black. Comparison of the sea ice maximal extent to the mixed layer depth shown in Fig. S1 (for the OBS case) while the black contour in the current plot indicates that deep convection is just off the outer limit of the sea ice maximum. The strong zonality of the Gulf stream is readily visible in the salinity field. The eddy resolving/permitting nature of the model configuration is evident in the plotted salinity colour bands. Inset panel shows each of the averaging regions highlighted with red for the Labrador Sea region, blue for the GIN Seas region, and yellow for the North Atlantic deep-water formation region.

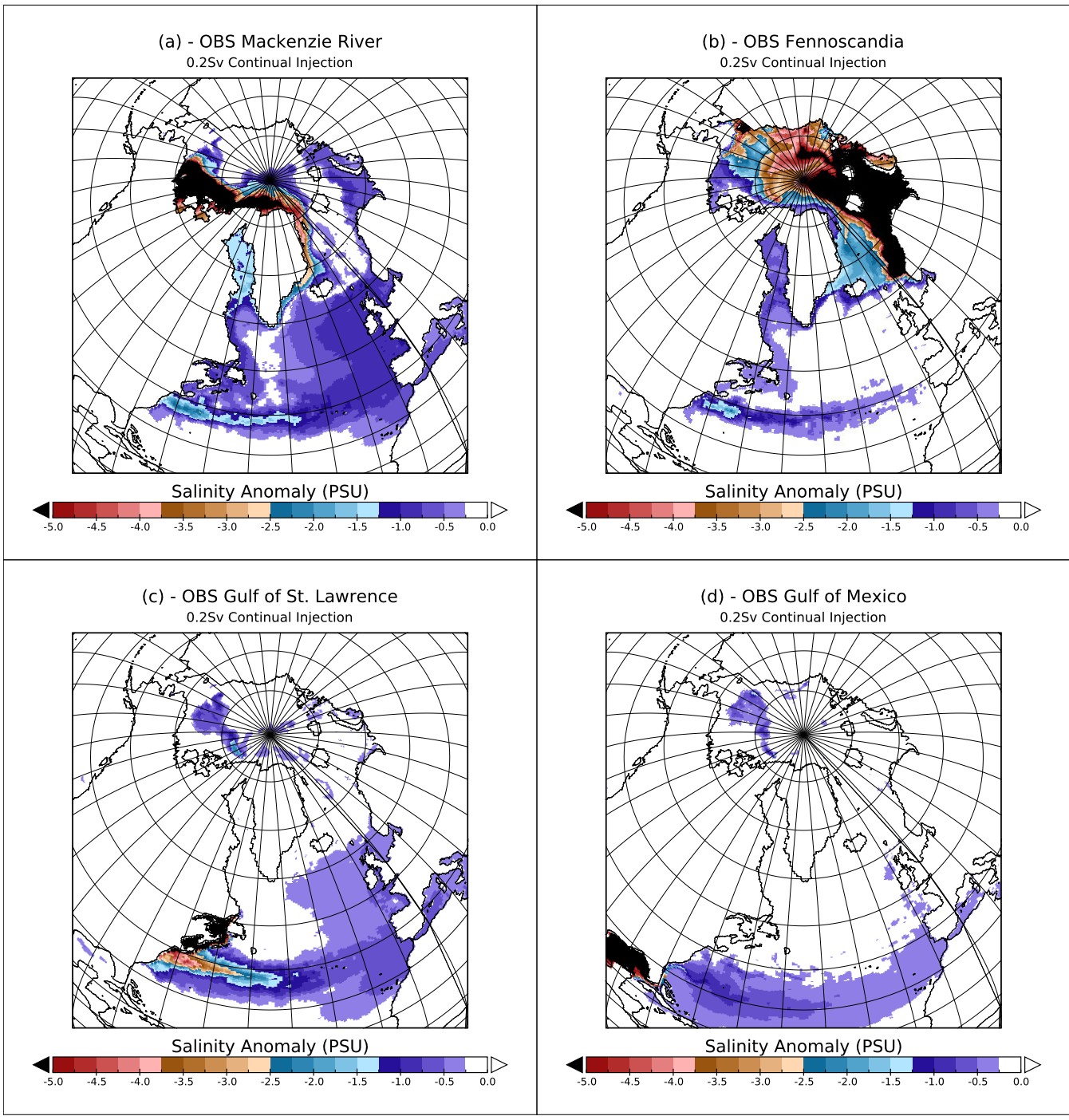

**Figure 3.** Time averaged sea surface salinity anomaly from the last 5 years of each of the OBS injection simulations, the CBS MAK experiment is available in the in Fig. S5.

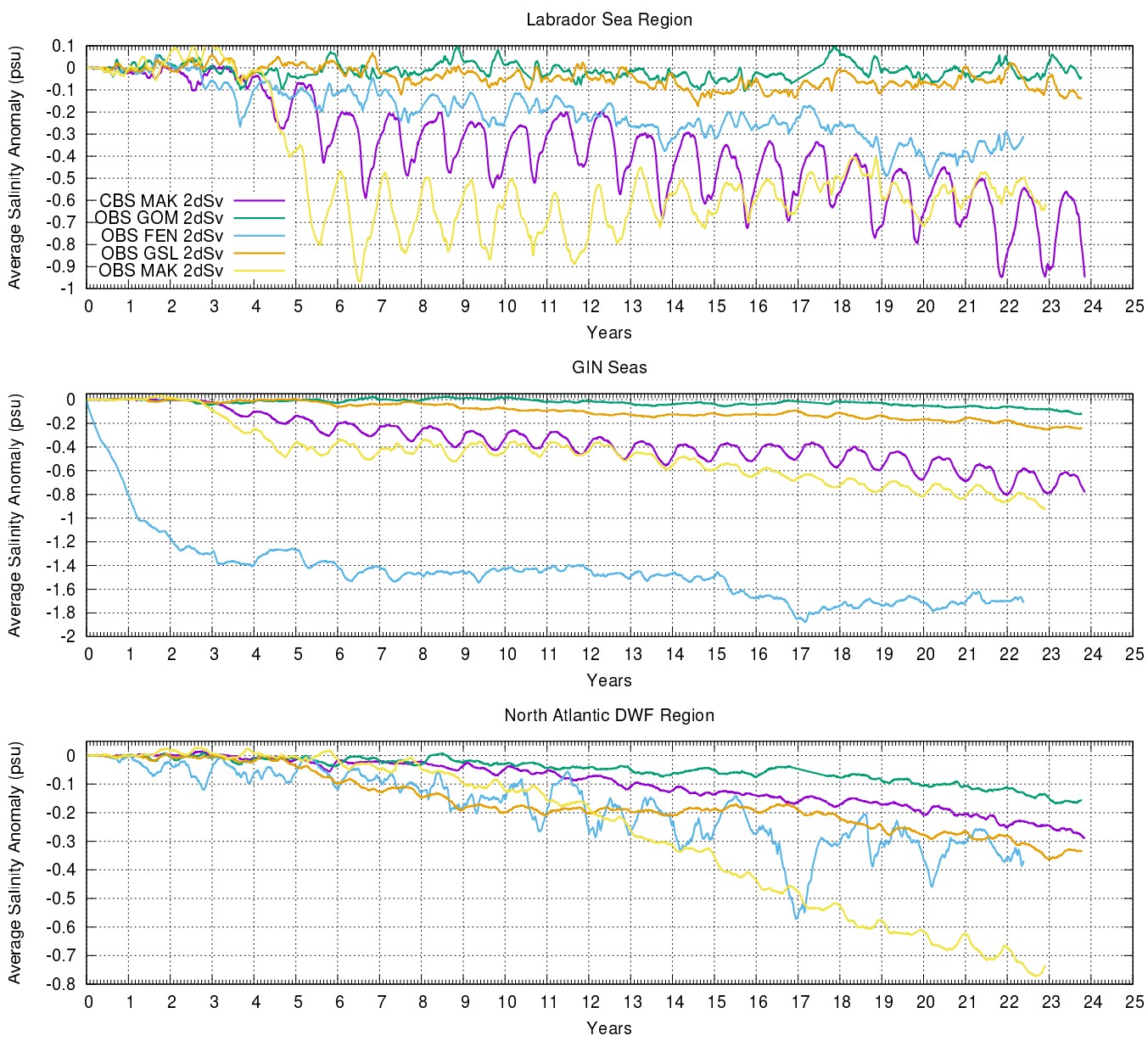

**Figure 4.** Sea surface salinity anomalies for each of our freshwater injection scenarios, calculated relative to their respective control runs. Each injection scenario uses 2dSv of freshwater continually injected at the location of their respective outlets. Each of the averaging regions is shown in the Fig. 2.

## 3.2 Freshwater transport paths

We begin our examination of the injection experiments by tracing the pathways of freshwater transport from each injection location. We present in Fig. 3 salinity anomalies at the surface for each of the four injection locations. Figures S6, S7, S8, and S9 show the salinity anomalies at 50m, 100m, and 150m depth. These anomalies are calculated as the differences between averages over the last 5 years of the injection experiments and the corresponding 5 years of the relevant control simulation. From Figures 3a, b, and c, it can be seen that freshwater from MAK, FEN, and GSL tends to follow a single, continuous pathway around the Arctic, into the East and then West Greenland Currents, and following the Labrador Current to the northern margin of the Gulf Stream. There, the freshwater accumulates at the separation point of the Gulf Stream from the eastern coast of North America and is advected eastward. At the western coast of Europe, the freshwater is mixed into the northern North Atlantic via the collapse of eddies. The main difference between different outlets along this pathway is the magnitude of freshwater at a given location, which is dependent on how far along this pathway the freshwater has traversed. The longer the pathway to that location, the greater opportunity for freshwater dilution through diffusion and eddy shedding and the greater time to get there. We note that if there is significant freshwater build-up along this pathway, it acts as a barrier, slowing down the transport of freshwater. This is exemplified in the case of the GSL in Fig. 3, which has much less freshwater in the eastern North Atlantic than the OBS MAK but much more entrainment of freshwater in the Gulf Stream at the location of its separation from the eastern coast of North America on the western North Atlantic.

The vertical distribution of freshwater along this pathway can be discerned by examining the path traced by freshwater injected at the mouth of the MAK when the Bering Strait is closed as shown in Fig. S5. Due likely in good part to the lack of wind stirring given perennial sea ice cover, the bulk of the salinity anomaly is located at the surface. As it fills the Canada basin, passes through Fram Strait along the continental shelf of Greenland, and joins the East Greenland Current, the freshwater remains primarily a surface signal. The concentration of freshwater in the surface current decreases dramatically, though, as it travels to the West Greenland Current and into the Labrador Sea and Baffin Bay. The large reduction in surface salinity anomaly along the east coast of Greenland coincides with the appearance of significant salinity anomalies at 100m depth and deeper. This is due to vertical mixing along the continental shelf of Greenland (with a local depth between 150-250m in this configuration) diluting the surface signature while introducing anomalies from the surface to 200m depth.

While the freshwater pathway when the Bering Strait is open in Fig. 3A and Fig. S6 is broadly similar to that when it is closed, there are distinct features that provide insight into the mixing and transport processes occurring in a glacial ocean. Firstly, the Arctic surface salinity anomaly does not spread into the Canada Basin to the same degree, because it is constrained to lie between the Transpolar Current (not noticeably present in the CBS case) and the coast of the Canadian Archipelago. As a result, the surface freshwater concentrations carried along the East Greenland Current and West Greenland Current and into the Labrador Sea are much stronger than when the Bering Strait is closed. However, it is unclear that this contrast would persist if the simulation and injection were long enough to saturate accessible Arctic Ocean sectors. When comparing against OBS/CBS results from Karami et al. (2021) (with the closed Canadian Arctic Archipelago) it is apparent that this contrast is present even in unforced simulations. With an OBS there is less spreading into the subpolar gyre region and the Gulf Stream. When the

Bering Strait is open there is a shift southward of the Gulf Stream and overall faster western boundary currents northwards and slower southwards of the Gulf Stream. Secondly, vertical mixing of the surface salinity anomaly appears to start earlier for the OBS case, in the shear zone of the Transpolar Current in the central Arctic. Thus, there is a stronger salinity anomaly at all depths to 150m off the northern coast of Greenland for the OBS case before becoming comparable to one-another along the
eastern coast. The primary differences in this pathway from what is observed and simulated for the present is that freshwater sourced from the MAK tends to flow eastward along the coastal margin into the Canadian Archipelago (Fichot et al., 2013; Condron and Winsor, 2012), which is closed at the time of the Younger Dryas.

Freshwater from FEN tends to follow two different routes to DWF regions in Fig. 3B and Fig. S7. One freshwater mass travels directly across the GIN Seas to eastern Greenland, following the surface currents shown in Fig. S3. The second water
mass is initially entrained in the Norwegian current, which carries the freshwater from the injection region northward to flood the Barents-Kara sea. The freshwater then circles around the Arctic basin before being transported southwards back into the GIN Seas and the North Atlantic via the East Greenland Current, following a similar pathway to the MAK. Similarly to the MAK experiments, the freshwater from FEN remains mostly in the top 50m of the water column until it reaches the continental margins of Greenland.

The freshwater from the GSL (Fig. 3C and Fig. S8) gets entrained in the Gulf Stream and only spreads meridionally on the eastern side of the North Atlantic, where it also becomes mixed vertically as it passes over the shallow (200-300m depth) continental margins. As previously noted, the Gulf Stream in our simulations is more zonal than during present day. Given conditions during the Bølling-Allerød were closer to present day rather than full glacial, our configuration may not be an accurate representation of the Gulf Stream just prior to the Younger Dryas. A more modern Gulf Stream, where surface
currents are more north-eastward, should result in greater freshwater transport to the North Atlantic DWF zone and GIN Seas, though again with substantial mixing.

Finally, freshwater released into the GOM initially fills that basin before leaking over the Florida shelf and into the Atlantic (see Fig. 3D and Fig. S9). Inflow from the Yucatán Channel acts as a barrier to the freshwater that has filled the GOM, preventing it from expanding southward. The lower sea level around the Younger Dryas results in a more isolated GOM
relative to present and helps to sequester the GOM from the Atlantic. As in the other scenarios, the freshwater remains in the uppermost layers as it passes over the Florida shelf. Afterwards, it mixes downward as it travels north and eventually becomes entrained in the Gulf Stream with freshwater present at least 200m deep. In neither the GSL nor the GOM injection is there evidence that the freshwater anomaly is able to cross the Gulf Stream as in Condron and Winsor (2012). The zonality of the Gulf Stream in this configuration does not affect this conclusion substantially. When the GOM experiment is repeated using
modern wind forcing from ERA40 (Uppala et al., 2005) in section S5, the Gulf Stream is less zonal and closer to present-day observations. Yet, the majority of the freshwater remains primarily in a zonal band, while detectable pockets of freshwater now enter the subpolar North Atlantic. Finally, none of these simulations account for the effect of sediment in the glacial runoff which can lead to bottom-riding (hyperpycnal) flow in (Parsons et al., 2001). Tarasov and Peltier (2005) suggested that outflow from the Mississippi (GOM) and the GSL at the magnitudes examined here would be heavily laden with sediment, rendering

the outflow hyperpycnal with a resultant change in transport. By comparison, the MAK basin has limited surface sediments, and freshwater outflow would be much less affected by this process.

## 3.3   Injection site impact on DWF region salinity

Having traced the pathways of injected freshwater from each outlet, we now examine their respective contributions (Fig. 4) to three potential DWF regions: the Labrador Sea, the GIN Seas and the northern North Atlantic. Labrador Sea salinity is
most strongly affected by freshwater injected into the MAK outlet. When the Bering strait is open, the peak freshening occurs within 7 years and appears to saturate after 10-15 years. In contrast, closing the Bering Strait reduces the freshening effect to half for the first decade of injection, after which its salinity anomaly gradually surpasses the OBS MAK. The FEN injection generates the next strongest anomaly in the Labrador Sea relative to the MAK outlets. None of the other tested outlets contribute noticeably to salinity anomalies in the Labrador Sea.

The GIN Seas region is most significantly affected by freshwater from the FEN injection, whose salinity anomaly is more than two times larger than that from the next largest contributor, the MAK. The reason for the importance of the FEN injection to GIN Seas salinity is largely due to its being within the averaging domain combined with the local ocean circulation directing FEN freshwater across the region.

Finally, the primary location of deep mixing in these simulations, the northern North Atlantic, is affected by injection into
all of the outlets examined here. The strongest contribution is from the OBS MAK injection, which generates approximately twice the freshening of the next strongest outlets, the FEN and GSL. Notably, the salinity anomaly from the FEN injection location exhibits a much larger seasonal cycle compared to that of the other tested outlets.The GSL's freshening effect appears to increase in a step-wise fashion. None of the simulations appear to have reached equilibrium in the North Atlantic, with the exception of FEN. The prominent seasonal cycle of FEN, which exhibits the largest amount of variability on inter-annual
timescales, reduces confidence in this conclusion. The CBS Mackenzie sourced simulation has a more delayed response compared to that of the corresponding OBS simulation, as expected with the enhanced boundary currents observed with an OBS, and it never reaches the rate of freshening achieved by the OBS over the duration of these simulations. Note that there is a detectable contribution to the salinity anomaly of the northern North Atlantic region from the GOM, although the salinity signal is not large enough in any single grid cell to be detectable in Fig. S9. Of all our explored injection scenarios, the GOM
scenario has the least impact with regards to salinity change in key DWF regions. The Mississippi River (primary meltwater drainage route to the GOM) therefore offers a possible escape valve for minimizing the impact of terrestrial meltwater injection on DWF, and therefore AMOC, at least on inter-annual to decadal timescales.

For comparison to conventional hosing studies, an order-of-magnitude calculation of the freshening effect of a 1-year 2dSv flux injected into each of the DWF regions (indicated in Fig. 2) is worth consideration. We assume that the freshwater displaces
existing seawater from the regions, that the injection region is evenly inundated with freshwater, and the freshwater is evenly mixed over the top 50m of the water column. We do not account for the eventual flow of water in or out of the regions. For further details see Section S6. Using the salinity field from the control run as our initial state, hosing directly onto the Labrador Sea region would result in a $-4.2$PSU change in salinity, which is more than 4x stronger a freshening effect than any of our

equilibrated injection runs. Hosing the GIN Seas region results in a $-0.65$PSU salinity change, which is very similar to the top layer salinity shown in Fig. 4 after 1 year of injecting into the FEN injection location (located within the GIN Seas region). Finally, hosing in the North Atlantic DWF region results in a $-1.26$PSU salinity change. As in the Labrador Sea region, this represents an approximately 2-4x larger change than observed from any of the injection experiments presented herein.

Our results have significant differences compared to those of Condron and Winsor (2012) and Hill and Condron (2014), which both imposed a large 50dSv flux of freshwater for only the first year of the simulations. The much larger rate of freshwater injection in those studies generated much greater mixing at the boundaries of the coastal boundary currents and led to a greater spreading of the freshwater in the Arctic and Atlantic oceans. Also, the freshwater in Condron and Winsor (2012) and Hill and Condron (2014) is readily transported across the Gulf Stream, routing freshwater from either the GSL or the Hudson Strait to south of Cape Hatteras and into the GOM and vice versa. An examination of the freshwater distributions in Figs. S5 and S6 shows none of the overall flooding of the North Atlantic that is present in Condron and Winsor (2012). The lower but continual flux in the simulations shown here also does not allow freshwater to penetrate the Gulf Stream as effectively by comparison to Condron and Winsor (2012). Indeed, results from Condron and Hill (2021) indicate that our freshwater flux is not of sufficient magnitude to allow it to penetrate the Gulf Stream, which would require at least a 10 fold increase in flux. As such, the GSL injection delivers, relatively, significantly more freshwater to the GIN Seas and North Atlantic DWF region than the GSL run in Condron and Winsor (2012) despite both a much lower flux and overall volume of freshwater.

Additionally, we can compare our results to Roche et al. (2009) and Lohmann et al. (2020). The former performed a wide suite of injection experiments using a much lower-resolution model with varying freshwater flux and injection location under LGM boundary conditions. The latter performed a set of 4 injection experiments using a model with enhanced grid resolution over large regions in the Arctic ocean and around the coasts while having the interior Atlantic grid resolution range upwards of 140km via their unstructured mesh approach. Since Roche et al. (2009) did not discuss the salinity signals at DWF sites directly, we interpret the freshening of the GIN Seas and northern North Atlantic DWF regions in this study to be analogous to changes in NADW export from their two sites of DWF. Our results are in broad agreement with both Roche et al. (2009) and Lohmann et al. (2020) except with regards to freshwater injected into the GOM. Roche et al. (2009) found their GOM injection to generate comparable or greater effects on NADW export than injection from the GSL. The freshwater signal at DWF sites in Lohmann et al. (2020) from GOM was also stronger than in this study. We attribute both differences to the lower resolutions of the Florida Strait and Gulf Stream in those studies (approximately 18 times coarser in Roche et al. (2009) and $\approx 2-3$x coarser in Lohmann et al., 2020). This lower resolution combined with the much longer duration of simulations in these studies would increase export rates and allow freshwater built up in the GOM time to flow out of the region and freshen the Atlantic. Finally, the higher resolution of the Gulf Stream in our set of simulations appears to make it a more effective barrier to freshwater transport than in either of these studies.

## 3.4 Implications for last deglacial interval

We now present the implications of our results to the routing of runoff during the deglacial interval. The glaciologically-modelled discharge time series from Tarasov and Peltier (2006) indicates there is a steady background discharge from NH ice

sheets into the oceans from before Heinrich Event 1 to MWP1A, largely from the Gulf of Mexico. During this time interval, we would expect extensive sea ice over the GIN and Labrador Seas, and thus deep-water formation regions largely aligning with the NADW region in Fig. 2. As such, the runoff from Fennoscandia should have had a continual suppressing effect on deep-water formation during this time interval relative to present-day conditions. Examining the RSL data, we see there is almost no change in sea level during this time interval indicating that the freshwater flux is largely in balance with NH ice sheet changes (1dSv over 1000 years contributes $\approx 8.8$m of eustatic sea level rise).

During the deglacial interval, the first major freshwater flux event is Heinrich Event 1 which is not well represented by our investigation that examines only liquid freshwater. However, we can make some inferences by combining our results with the iceberg modelling results of Hill and Condron (2014); Condron and Hill (2021). Examining the iceberg distributions from these studies we can conclude that the resulting iceberg meltwater distribution would be most similar to our Gulf of St. Lawrence outlet experiments, and as such would have one of the larger impacts upon deep-water formation in the North Atlantic. The first major liquid freshwater event we can discuss in the context of our results is MWP1A. It occurs early during the Bølling-Allerød interval, which is associated with rapidly increasing central Greenland temperatures, and increased freshwater discharge to the North Atlantic and brief century-scale minor reductions in AMOC (Obbink et al., 2010). As MWP1A occurred during the near-interglacial climate state of the Bølling, DWF likely occurred in the GIN Seas. The reconstruction presented in Tarasov and Peltier (2006) largely attributes MWP1A runoff to the Mississippi River along with discharge into the Atlantic (from Newfoundland to Florida, with discharge via the Hudson River in contemporary New York State dominating). Paleoceanographic records from Orca Basin adjacent to the mouth of the Mississippi show the strongest deglacial freshening during this time interval (Broecker et al., 1989). In contrast, some studies (such as Gregoire et al., 2016) appear to implicate freshwater routed through the Mackenzie River for MWP1A, whereby saddle collapse between the Cordilleran and Laurentide ice sheets is considered as the main mechanism for the rapid increase in outflow and sea level. As we have demonstrated, freshwater routed through the Mackenzie River is expected to have a stronger impact on freshening sites of deep-water formation relative to the Mississippi River, at least on decadal timescales. Similarly, the GSL (and very likely outlets south thereof, such as the Hudson River, Pendleton et al., 2021) also have relatively low impact on salinity over the GIN Seas. Therefore, a dominant North American freshwater discharge from the GSL and outlets south thereof, along with minimal freshwater discharge from the Mackenzie River and Fennoscandia may help to explain the lack of a strong climate system response to the enhanced freshwater input into the ocean during MWP1A.

However, this conclusion has to be understood in the context of the assumptions that went into these results, particularly the treatment of freshwater as an external forcing to the system and not part of a coupled ice-ocean-atmosphere system. It is worth noting that there is no evidence B-A warming occurred in response to a cessation of freshwater forcing (Tarasov and Peltier, 2006), as it was reproduced by in Liu et al. (2009), for example. If the B-A warming occurred as a manifestation of internally driven, Dansgaard-Oeschger-like variability, then it may be that the mechanisms that led to this warming could have also stabilized the AMOC against the freshwater resulting from the warming. It may be a similar mechanism also operated at the transition out of the Y-D and into the Holocene, when runoff is reconstructed to have derived from the GSL, which effectively reached North Atlantic deepwater formation sites in this study. Discerning whether this is indeed possible would

require an analysis of the impact of freshwater on spontaneous Dansgaard-Oeschger-like variability, which we leave to future work.

Combining previous work highlighting the importance of glacial runoff from the MAK (Condron and Winsor, 2012; Keigwin et al., 2018), the impact of freshwater from FEN presented in Toucanne et al. (2009), and the results we present here, we conclude that the most plausible sources of glacial runoff to cause rapid cooling, while minimally impacting RSL, would be the Mackenzie River or Fennoscandia. The discharge data presented in Tarasov and Peltier (2006) supports this conclusion with respect to the Mackenzie River outlet. However, our results do not preclude non-linear effects from a short-lived freshwater discharge event from other outlets, such as a large flux/flooding event over a very short time interval (i.e. timescales below the resolution of studies such as Tarasov and Peltier, 2006) as discussed by Broecker (2006); Teller et al. (2002). Nor does our work address a potential flush of freshwater from anywhere it has accumulated, such as in Baffin Bay for the MAK injections.

## 4   Conclusions

This study provides the first assessment of freshwater transport to deepwater formation regions under Younger Dryas conditions using realistic freshwater injection amounts applied to a range of plausible source regions in a suite of eddy-permitting simulations. We have addressed three main shortcomings in common practice for freshwater injection experiments that inflate the salinity anomalies at locations of DWF. The first shortcoming we address is the injection of freshwater directly over the locations of DWF rather than at its source location to mitigate unresolved $O(< 50\text{km})$ oceanic processes known to be important in the transport of glacial runoff. Using our model configuration, we find the transport of freshwater from the coast to sites of deepwater formation leads to a reduction in the effective freshwater forcing. We find in this study that one year of 2dSv injection at the mouth of the MAK (CBS) yields a freshening equivalent to direct regional hosing by amounts of $\approx 0.31$dSv in the Labrador Sea, $\approx 0.33$dSv in the northern North Atlantic, and $\approx 1.85$dSv in the GIN Seas (using the same method and simplifications as in Section 3). Thus, while these practices may mitigate the inability of coarse resolution models to adequately resolve the small-scale features that are key to freshwater transport, like boundary currents and mesoscale eddies, applying 2dSv directly into these regions is an inaccurate representation of the transport processes involved. Since non-eddy-permitting models are currently and will likely continue to be used for paleoclimate studies, we are presently exploring better ways to mitigate this problem, the results of which are the subject of an upcoming study and outside the scope of this work.

The second shortcoming is the use of unrealistically large freshwater amounts. We find that limiting freshwater amounts to glaciologically-constrained values results in less diffusive spreading of the freshwater across the North Atlantic. In addition, the lower amounts are unable to traverse the Gulf Stream, isolating the salinity anomalies introduced north and south of the Gulf Stream.

The final shortcoming involves the use of present-day rather than paleo-bathymetry, and especially its effect on the Bering Strait. For the most proximal site to the Bering Strait, the Mackenzie River, we find that the opening of this gateway leads to a faster increase of freshwater export from the Arctic ocean and a larger downstream effect on the salinity of the northern North Atlantic.

We characterize which injection source region has the strongest freshening effect at three different potential deepwater formation (DWF) regions, the Labrador Sea, GIN Seas and northern North Atlantic. For DWF in the northern North Atlantic (most commonly occurring when the climate is in a glacial state with extensive sea ice Braconnot et al., 2011), freshwater introduced into the MAK outlet with an OBS imposes the largest freshening effect. Yet, we detect significant freshening from all injection outlets. For intermediate and DWF in the Labrador Sea, freshwater from the MAK generates the greatest freshening, with FEN having the next largest impact. Opening the Bering Strait approximately doubles the rate of freshening over the first 10 years of MAK injection. For GIN Seas DWF, freshwater from FEN is the strongest contributor to salinity anomalies. The implications of these results to our overarching question of how significant RSL changes, such as occurred during MWP1A, could occur without a consequent effect on DWF rates and climate is that the majority of freshwater from North American Ice Sheets needs to enter the ocean south of the Gulf Stream (along the coast of North America) to minimally impact sites of DWF. As well, the reduction in meridional transport of freshwater across the Gulf Stream observed in our results is a feature which ought to be equally applicable, and considered when not explicitly resolved, for both paleo-climate and future-climate investigations.

Finally, our results raise two questions which we leave to future work. Could a build-up and subsequent flushing (via changing oceanic gateways or changes in perennial sea ice) of freshwater in a partially isolated region, such as Baffin Bay, lead to a delayed onset of cooling after a change in routing or increase in glacial runoff? Additionally, can a transition from a stadial to an interstadial climate provide some means of stabilizing AMOC to the effects of freshwater and thus allow for both increased glacial runoff and increased warming such as seen at the onset of the Bølling-Allerød?

*Data availability.* Model output data is available upon reasonable request.

*Author contributions.* All authors assisted with experimental design and analysis. RL prepared the manuscript with contributions from all authors.

*Competing interests.* The authors declare that they have no conflict of interest.

*Acknowledgements.* The authors would like to thank the anonymous reviewer and Pasha Karami for their assistance in improving this manuscript during the review process. The authors would also like to thank those at the GNU and Fedora projects,Kernel.org and in particular those responsible for GNU Parallel (Tange, 2011) whose software greatly sped up and streamlined the analysis in this work. This research was enabled in part by support provided by SciNet (www.scinethpc.ca) and Compute Canada (www.computecanada.ca) through both Resources for Research Groups allocations and the Rapid Access Service. This is a contribution to the ArcTrain program, which was supported by the

Natural Sciences and Engineering Research Council of Canada. HA was funded by the German Federal Ministry of Education and Research (BMBF) as a Research for Sustainability initiative (FONA) through the project PalMod.

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
