# Peer review of "Freshwater Routing In Eddy-permitting Simulations Of The Last Deglacial : Impact Of Realistic Freshwater Discharge"

_Climate of the Past, 2021_

## Referee Comment (RC2)

**Review: Eddy permitting simulations of freshwater injection from major Northern Hemisphere outlets during the last deglacial, by Love et al.**

The manuscript explores freshwater transport to deep-water formation regions under Younger Dryas conditions by using high resolution global ocean model simulations and with more realistic (compared to previous studies) freshwater injection amounts.

I think it is an interesting work to be published in Climate of the Past. My major problem with the manuscript is about the presentation of the results (text, structure and figures) which makes the readers to use their imaginations instead of the paper. The results do not really convey the main message and some important points are not discussed. The authors should re-structure the text and re-write some parts to ease the reading, and put some figures from the supplementary to the main paper. Below, some of my major and minor comments regarding the text and analysis are given in more details:

**Major comments:**

Introduction:
-The main story and motivation of the study is hidden behind all the text. It should be re-written with a clear and standard structure, where general description of the problem, goal of the paper and previous studies are clearly discussed.

-Figure 1, is a very nice figure but it is not really discussed and not clear why you used it. I think it deserves more explanation.

Experimental design:
-The model description should be improved (e.g., you need to clearly specify that you use a global model and how many vertical levels your model has). Next, discuss the forcing. Then, explain the initialization of experiments, the control runs, number of spin up years (exact numbers), and total simulation years. Last, explain the experiments with all the needed details. In the current version, you might have given most of these information but it is done in a messy way.

-I think your Figure S2 should be discussed in this section and be used as a main figure and not a supplementary.

Results:

-I wonder why there is no summer sea ice in the Arctic in the region above (north of) Greenland? As far as I know that is a region that is covered by sea ice in summer (for present day condition).

-How is the Gulf Stream (that is found to be highly zonal), sea ice and regions of deep-water formation in this study compared to previous studies?

-Figure S5: you show only 1 year (the last year of simulation). Please choose similar intervals and same number of simulation years for all the figures (e.g., 5-y mean of year x to y). Same for Figure 2, what is meant by single day? For your study, yearly-mean values should be fine but take the average over several simulation years.

-Would be interesting to see the timeseries plot for the MLD in Labrador and Nordic seas.

-Figure S6-AMOC:

You initialize the model from the experiment by Hill and Condron (2014), right? But why your experiment's AMOC is about 6 Sv at year -10 while it should be larger given the AMOC in Hill and Condron (2014)? If I am mistaken, please explain this part better.

Overall, the AMOC in your study seems to be smaller than some similar studies (https://agupubs.onlinelibrary.wiley.com/doi/pdf/10.1002/2015GL064583), right?

What is the difference in AMOC between CBS and OBS control runs? The AMOC difference between the experiments seems to be small (1 or 2 Sv), and perhaps within the range of model internal variability. I am not sure if the AMOC can give any conclusive view.

-Figure 4:

Is it surface salinity? Except in the middle panel, the salinity anomaly shows a downward trend in some of the experiments (for instance the CBS MAK). You need to be careful how you interpret these as the model is not clearly far from equilibrium.

-One implication of this study is for the Younger Dryas event which is linked to temperature changes, and I was expecting to see a plot for the sea surface temperature (SST). I realized that this an only ocean model study but would still be interesting to see the (indirect) impact of different FW injection on SST.

-I will include a figure similar to Figure 3 but for BS closed in the main paper. Also Figure S6 is better to be in the main paper.

**Minor comments:**

Title: needs to be adjusted. Is it really during the last deglacial?

Line 9: You are using paleo forcing and paleo-bathymetry, please correct it.

Line 35: Sv is the common unit to use, and dSv is not really helping to make things easier.

Line 91-92: Does the model really captures the coastal boundary currents?

Line 104: "The first…": revise

Line 134: "...discussed in Experimental design section": is not discussed

Figure S6: It is strange to use negative time values for the model spin up period.

---

## Community Comment (CC1)

**Review of: "Eddy permitting simulations of freshwater injection from major Northern Hemisphere outlets during the last deglacial"**

**By: Ryan Love, Heather Andres, Alan Condron, and Lev Tarasov**

**Reviewed by: [1]Jenny Jardine, [2]Pearse Buchanan, and 1 other**

**Institution: [1]National Oceanography Centre Liverpool and the [2]University of Liverpool**

**Description:** The authors investigate the impact of glacial freshwater runoff from realistic locations in the North Atlantic and Arctic in high-resolution, eddy-resolving simulations using the MITGCM ocean model under glacial boundary conditions. The locations of freshwater release include the Gulf of Mexico, the Gulf of St. Lawrence, a cite off Norway termed Fennoscandia, and the Mackenzie River in the high western Arctic. They conduct short transient simulations (decades) and observe the changes in salinity over different regions of the North Atlantic and the effect on the Atlantic Meridional Overturning Circulation, measured at 26°N to be consistent with contemporary measurements at the RAPID array. They find that freshwater released at realistic rates and locations has little effect on the AMOC, but that surface salinities are appreciably fresher in deep water formation regions when freshwater is released from Fennoscandia and the Mackenzie River.

**Main Comments:**

The major point of the paper is simple enough. Freshwater released in realistic locations, with realistic circulation and at realistic rates doesn't reduce deep water formation. This would suggest that despite glacial conditions and realistic release of freshwater during a strong meltwater pulse, the authors are showing that the AMOC has not reached a tipping-point for collapse in their model, which is an interesting result. However, the paper is somewhat difficult to read so we advocate for an improvement in the clarity of the paper. Clarity could be improved by improving the title, abstract and reporting of results. With respect to the title, it reads as methodology, not as a main finding. With respect to the abstract, we feel that it is much too long. With respect to the results, we feel that the important points are lost in very detailed reporting and confusing sentences.

Our main comment is therefore that the authors should improve the clarity of the manuscript. However, we also have some more specific comments too, some of which will likely be addressed by an improvement in overall clarity:

**Other Comments:**

- There are many acronyms in this paper. As it is a short paper, we think these should be spelled out more to ease the readability. https://www.natureindex.com/news-blog/science-research-papers-getting-harder-to-read-acronyms-jargon

Introduction:

- In the introduction, it is mentioned that there are "at least three common experimental design problems", but the authors only expand on two.

- It would be nice to have some information in the introduction for why the authors chose the sites they chose. Are these areas known to be the major outflows of freshwater during the glacial? Are there others that are not accounted for?
- It would also be nice for the introduction to talk more to why an AMOC collapse is thought to have occurred many times in the past. It is implied, but not clearly stated in the introduction. It is also not stated why we might be interested in AMOC collapse today, which may be obvious to the authors but would be worth stating.
- Another topic that is not mentioned is the bistability of the AMOC. There has been much work on the existence of "tipping-points" (e.g. most recently Lohmann & Ditlevsen, 2021, PNAS), whereby over a certain threshold of freshwater hosing the AMOC collapses, but underneath that threshold it does not. This is an important concept to include given that despite some significant freshening in your experiments there is little effect on the AMOC.

Results/Discussion:
- The ice extents in km$^2$ are quite low given that the record minimum in 2017 was 14.3 million km$^2$.
- Line 170 should read ". . down to 200 m depth. . ."  [This comes of too much use by scientists of "high" instead of "large"]
- Line 186. Direct transport from FEN across the GIN seas is not clear in the figure, the proportion of fresh water shown is very small.
- Line 191 becomes clear looking at the figure but there has to be meridional transport to get from the Gulf of St. Lawrence to the Gulf Stream and it is curious to read "eastern . . North Atlantic"
- Lines 229-231. A curious statement. As though the meltwater is trying not to affect AMOC.
- It should also be noted that the authors did not complete a combined experiment where all sites received increased freshwater fluxes at the same time. This might have been sufficient to tip the AMOC into a collapsed state. At the very least, this should be discussed. At most, another simulation should be performed with all four release sites simultaneously releasing freshwater.

Figures:
- We suggest a change in the colour scheme of figures from jet to something more colour-blind friendly.
- Figure 1 is confusing and needs more details. We do not know how to interpret it.

---

## Author Response (AR1)

**RC1 Referee Comment:**

This paper presents a set of sensitivity experiments using an ocean-only GCM forced with atmospheric boundary conditions from CCSM3 in a LGM configuration. The ocean model has a relatively high resolution (about 5 times greater than most CMIP5 models) and can be considered as eddy permitting. The authors perform a set of 5 simulations where freshwater is released from different locations that corresponds to potential region of glacial icebergs and meltwater discharges in the last ice age and deglaciation. All the sensitivity simulations are shorter than 20 years, and the focus is put on the pathway of the freshwater in the Atlantic Ocean, and its impact on the regions of oceanic deep convection. It is found that freshwater pathways are highly dependent on the release location, as was already highlighted in a few studies.

The scientific topic tackled by this study is of interested, given the very large uncertainty concerning freshwater pathway that might be related with oceanic resolution (e.g. Gillard et al. 2016). These pathways are clearly of importance concerning the response of the deep convection and AMOC as highlighted in e.g. in Swingedouw et al. (2013). The novelty of the study as compared to existing work by e.g. Condron &Winsor (2012) is the use of glacial boundary conditions, and a more systematic analysis of the different potential outlet locations as well as the consideration of smaller rate of freshwater release, more in line with recent reconstructions. The pathways of freshwater release are of great importance to refine our understanding concerning the last deglaciation notably and the impact of freshwater release by melting ice sheet.

Nevertheless, there are a number of major caveats which might strongly limit the utility of these experiments

As highlighted by the authors, the length of their simulation is very short, which strongly hamper the interpretations of the results from those simulations for paleoclimate timescales, which are usually two order magnitude longer, as illustrated in Fig. S1 from the paper
The use of glacial boundary conditions apparently lead to a collapse of the AMOC in the ocean-only GCM used. The authors qualified it as a glacial state, but Fig. 6 shows a weakening AMOC index in the control simulation (which is thus not equilibrated at all) towards values of 2-4 Sv that rather correspond to an off state than a weak glacial states, according to e.g. Ganapolski and Rahmtorf (2001). AOGCMs indeed do not produce such weak state in glacial condition (e.g Kageyama et al. 2013, with all AOGCMs showing value larger than 5 Sv in their mean state). Considering an off state has major implications in term of barotropic circulation, notably in the subpolar gyre, which makes the relevance of those results doubtful for examining freshwater pathways at the beginning of e.g. the Younger Dryas as it is suggested in the paper.
The updates with former work is quite far from substantial, and the models used is very close to the one used in e.g. Condron & Winsor (2012). After almost a decade, computing power have strongly increased, so that these simulations now cannot be really considered state-of-the-art anymore, since far higher-resolution ocean-only simulations now exist, and do show that having even stronger resolution play a crucial role for the mean state of the AMOC (cf. Hirschi et al. 2020). As such, I am surprised that the authors still consider such short simulations (cf. point 1). Improvements in our understanding of the impact of ocean resolution from models of oceanic circulation need to be more appropriately discussed (cf. Hirschi et al. 2020, Le Corre et al. 2020)
The discussion of the implications of their results for paleoclimate understanding is very weak and deserve to be strengthen. What does those results mean in regard to existing literature that GOM and GSL affect so weakly the convection zones? What does that mean in terms of last deglaciation storylines?

While the first caveat is discussed appropriately in the paper, the 3 others are very poorly covered, if not at all. I therefore cannot recommend the paper to publication until appropriate discussions of these caveats are provided.

Please find below some specific points that provide further insights on the 4 main points listed above.

Figure 1: The data are difficult to see during YD due to very strong red. Please consider another colour to allow proper examination of the curves

Line 41 and elsewhere: "eg." Should be replaced by "e.g."

Line 90: such ocean-only model are simulations are not that costly within present-day computing time standard (e.g. Penduff et al. 2018 who considered 50 members of multi-decadal high resolution simulations…). Improvements as compared to former work with Condron as co-author, dating than almost a decade is not clear at all, while the main message remains also quite similar with this former work.

Line 89-90: How many vertical levels in the model?

Line 99-100: more should be said concerning the experimental design. Since these are ocean-only simulations, how are considered the boundary conditions? Is there any SSS restoring? How evaporation fluxes are computed,

Line 134: this very zonal Gulf Stream might also be related with the fact that the AMOC is in an off-state, since this can strongly impact Gulf Stream pathway (e.g. Caesar et al. 2018)

Line 141: "Labrador" Sea (not sea)

Line 145-146: This claim is not supported by anu figures, and I strongly doubt of this, given the very small value at 26°N. The AMOC is state rather resemble an off-state. Can we see the meridional streamfunction in the last 10 years of the control simulation?

Line 152: "glacial mode" sounds very optimistic. The authors might need to discuss more what is known from data and models concerning the mean state of the AMOC during the LGM…

Line 153: "reasonable". This might be a bit too much optimistic as well I think. Please discuss appropriately the state of your AMOC, or provide more evidences to support that it can be considered as a glacial state.

Line 225: "yr" is not defined.

line 267: A proper discussion of the implications in terms of the storyline of AMOC changes over the deglaciation and the link with freshwater release should be provided. As an example, we can assume that those experiments strongly support a major role for freshwater release from Fennoscandia, as suggested in e.g. Toucanne et al. 2009. Please, further elaborate on this topic in light of existing literature.

Line 267: An additional caveat is not properly discussed which is the fact that the authors consider here ocean-only model, which prevent from considering any potential coupled ocean-atmosphere feedback, which might play a role.

Line 268: "under Younger Dryas conditions": this statement does not really reflect the off state that is simulated in the control simulation.

Line 276-279: it is quite unclear from where those estimates come from, which is weird to provide in the conclusion, since not shown in the result section. I assume, they are estimated from a similar approach as in line 232-241, which is considering an ocean without any circulation at all. This is quite a strong hypothesis… Thus, I'm not sure those estimates are really useful, especially in the conclusion.

Line 283: "better ways to mitigate this problem": this sentence is quite enigmatic. Can you please clarify what is meant here?

Line 284-285: it should be stated here that these investigations are done in an off-state for the AMOC, and during only 20 years.

Additional references

Caesar, L., Rahmstorf, S., Robinson, A., Feulner, G., Saba, V., 2018. Observed fingerprint of a weakening Atlantic Ocean overturning circulation. Nature 556, 191–196. https://doi.org/10.1038/s41586-018-0006-5

Ganopolski, A. and Rahmstorf, S.: Rapid changes of glacial climate simulated in a coupled climate model, Nature, 409, 153–158, 2001.

Gillard LC, Hu X, Myers PG, Bamber JL (2016) Meltwater pathways from marine terminating glaciers of the Greenland ice sheet. Geophysical Research Letters 43(20):10,873{10,882, DOI:10.1002/2016GL070969

Penduff, T., G. Sérazin, S. Leroux, S. Close, J.-M. Molines, B. Barnier, L. Bessières, L. Terray, and G. Maze. (2018). Chaotic variability of ocean heat content: Climate-relevant features and observational implications. Oceanography 31(2):63–71, https://doi.org/10.5670/oceanog.2018.210.

Kageyama M., et al. (2013) Climatic impacts of fresh water hosing under Last Glacial Maximum conditions: a multi-model study Climate of the past, 9, 935-953, 2013. doi:10.5194/cp-9-935-2013.

Le Corre, M., Gula, J., Tréguier, A.-M., 2020. Barotropic vorticity balance of the North Atlantic subpolar gyre in an eddy-resolving model. Ocean Sci. 16, 451–468. https://doi.org/10.5194/os-16-451-2020

Swingedouw D., Rodehacke C., Behrens E., Menary M., Olsen S., Gao Y., Mikolajewicz U., Mignot J., Biastoch A. (2013) Decadal fingerprints of fresh water discharge around Greenland in a multi-models ensemble. Climate Dynamics 41, pp 695-720, DOI: 10.1007/s00382-012-1479-9

Toucanne et al. (2009) Timing of massive 'Fleuve Manche' discharges over the last 350 kyr: insights into the European ice-sheet oscillations and the European drainage network from MIS 10 to 2. Quaternary Science Reviews 28 (13-14), pp. 1238-1256

**RC2 Referee Comment:**

Review: Eddy permitting simulations of freshwater injection from major Northern Hemisphere outlets during the last deglacial, by Love et al.

The manuscript explores freshwater transport to deep-water formation regions under Younger Dryas conditions by using high resolution global ocean model simulations and with more realistic (compared to previous studies) freshwater injection amounts.

I think it is an interesting work to be published in Climate of the Past. My major problem with the manuscript is about the presentation of the results (text, structure and figures) which makes the readers to use their imaginations instead of the paper. The results do not really convey the main message and some important points are not discussed. The authors should re-structure the text and re-write some parts to ease the reading, and put some figures from the supplementary to the main paper. Below, some of my major and minor comments regarding the text and analysis are given in more details:

Major comments:

Introduction:
-The main story and motivation of the study is hidden behind all the text. It should be re-written with a clear and standard structure, where general description of the problem, goal of the paper and previous studies are clearly discussed.
-Figure 1, is a very nice figure but it is not really discussed and not clear why you used it. I think it deserves more explanation.

 Experimental design:
-The model description should be improved (e.g., you need to clearly specify that you use a global model and how many vertical levels your model has). Next, discuss the forcing. Then, explain the initialization of experiments, the control runs, number of spin up years (exact numbers), and total simulation years. Last, explain the experiments with all the needed details. In the current version, you might have given most of these information but it is done in a messy way.

-I think your Figure S2 should be discussed in this section and be used as a main figure and not a supplementary.

Results:
- I wonder why there is no summer sea ice in the Arctic in the region above (north of) Greenland? As far as I know that is a region that is covered by sea ice in summer (for present day condition).
- How is the Gulf Stream (that is found to be highly zonal), sea ice and regions of deep-water formation in this study compared to previous studies?
- Figure S5: you show only 1 year (the last year of simulation). Please choose similar intervals and same number of simulation years for all the figures (e.g., 5-y mean of year x to y). Same for Figure 2, what is meant by single day? For your study, yearly-mean values should be fine but take the average over several simulation years.
- Would be interesting to see the timeseries plot for the MLD in Labrador and Nordic seas.
- Figure S6-AMOC: You initialize the model from the experiment by Hill and Condron (2014), right? But why your experiment's AMOC is about 6 Sv at year -10 while it should be larger given the AMOC in Hill and Condron (2014)? If I am mistaken, please explain this part better. Overall, the AMOC in your study seems to be smaller than some similar studies (https://agupubs.onlinelibrary.wiley.com/doi/pdf/10.1002/2015GL064583),

right? What is the difference in AMOC between CBS and OBS control runs? The AMOC difference between the experiments seems to be small (1 or 2 Sv), and perhaps within the range of model internal variability. I am not sure if the AMOC can give any conclusive view.

-Figure 4: Is it surface salinity? Except in the middle panel, the salinity anomaly shows a downward trend in some of the experiments (for instance the CBS MAK). You need to be careful how you interpret these as the model is not clearly far from equilibrium.
-One implication of this study is for the Younger Dryas event which is linked to temperature changes, and I was expecting to see a plot for the sea surface temperature (SST). I realized that this an only ocean model study but would still be interesting to see the (indirect) impact of different FW injection on SST.

-I will include a figure similar to Figure 3 but for BS closed in the main paper. Also Figure S6 is better to be in the main paper.

Minor comments: Title: needs to be adjusted. Is it really during the last deglacial?
Line 9: You are using paleo forcing and paleo-bathymetry, please correct it.
Line 35: Sv is the common unit to use, and dSv is not really helping to make things easier.
Line 91-92: Does the model really captures the coastal boundary currents?
Line 104: "The first…": revise
Line 134: "...discussed in Experimental design section": is not discussed
Figure S6: It is strange to use negative time values for the model spin up period.

**CC1 Community Comment:**

Main Comments: The major point of the paper is simple enough. Freshwater released in realistic locations, with realistic circulation and at realistic rates doesn't reduce deep water formation. This would suggest that despite glacial conditions and realistic release of freshwater during a strong meltwater pulse, the authors are showing that the AMOC has not reached a tipping-point for collapse in their model, which is an interesting result. However, the paper is somewhat difficult to read so we advocate for an improvement in the clarity of the paper. Clarity could be improved by improving the title, abstract and reporting of results. With respect to the title, it reads as methodology, not as a main finding. With respect to the abstract, we feel that it is much too long. With respect to the results, we feel that the important points are lost in very detailed reporting and confusing sentences. Our main comment is therefore that the authors should improve the clarity of the manuscript. However, we also have some more specific comments too, some of which will likely be addressed by an improvement in overall clarity:

Other Comments:
• There are many acronyms in this paper. As it is a short paper, we think these should be spelled out more to ease the readability.
https://www.natureindex.com/newsblog/science-research-papers-getting-harder-to-read-acronyms-jargon
Introduction:
• In the introduction, it is mentioned that there are "at least three common experimental design problems", but the authors only expand on two.
• It would be nice to have some information in the introduction for why the authors chose the sites they chose. Are these areas known to be the major outflows of freshwater during the glacial? Are there others that are not accounted for?
• It would also be nice for the introduction to talk more to why an AMOC collapse is thought to have occurred many times in the past. It is implied, but not clearly stated in the introduction. It is also not stated why we might be interested in AMOC collapse today, which may be obvious to the authors but would be worth stating.
• Another topic that is not mentioned is the bistability of the AMOC. There has been much work on the existence of "tipping-points" (e.g. most recently Lohmann & Ditlevsen, 2021, PNAS), whereby over a certain threshold of freshwater hosing the AMOC collapses, but underneath that threshold it does not. This is an important concept to include given that despite some significant freshening in your experiments there is little effect on the AMOC.
Results/Discussion:
• The ice extents in km2 are quite low given that the record minimum in 2017 was 14.3 million km2 .
• Line 170 should read ". . down to 200 m depth. . ." [This comes of too much use by scientists of "high" instead of "large"]
• Line 186. Direct transport from FEN across the GIN seas is not clear in the figure, the proportion of fresh water shown is very small.
• Line 191 becomes clear looking at the figure but there has to be meridional transport to get from the Gulf of St. Lawrence to the Gulf Stream and it is curious to read "eastern . . North Atlantic"
• Lines 229-231. A curious statement. As though the  meltwater is trying not to affect AMOC.
• It should also be noted that the authors did not complete a combined experiment where all sites received increased freshwater fluxes at the same time. This might have been sufficient to tip the AMOC into a collapsed state. At the very least, this should be discussed. At most, another simulation should be performed with all four release sites simultaneously releasing freshwater.
Figures:
• We suggest a change in the colour scheme of figures from jet to something more colour-blind friendly.
• Figure 1 is confusing and needs more details. We do not know how to interpret it.

**Author Responses**

Line counts referenced by authors reflect the updated manuscript except where noted.

**RC1 Point-By-Point Reply:**

Firstly we would like to thank the anonymous reviewer for the time they have taken to read through the manuscript and make their suggestions. However we do have to refute several of their suggestions as they unfortunately seem to have mistaken the goals of our investigation, suggesting we may need to clarify these goals more explicitly in the manuscript in addition to adding additional content around caveats for the methods employed. An itemized set of responses to their comments is below, beginning with the more substantial points and technical points following.
* * *
*As highlighted by the authors, the length of their simulation is very short, which strongly hamper the interpretations of the results from those simulations for paleoclimate timescales, which are usually two order magnitude longer, as illustrated in Fig. S1 from the paper*
*...the first caveat is discussed appropriately in the paper..*

We agree that one limitation of our study is the short duration of the simulations in that there were still non-zero trends in some relevant climate metrics (i.e. AMOC). That is why we raise this point in the paper in a way that the reviewer describes as "appropriately discussed". We think it's also important to draw attention to the fact that the duration of the simulations is long enough for the surface transports of freshwater, which is the focus of our work, to have stabilized in most of the runs. Nevertheless, since receiving this review, we have resubmitted the runs, they have been extended by a few years (5-10 years run depending). Updated figures (Figs. 3 and 4) with these additional data and any updated discussion where required are included in the revised text. There have been no major changes to the discussion because of this extension as the relative ordering of the freshening has remained the same, excepting that for the North Atlantic DWF site the OBS Mackenzie River injection has become comparable to the Fennoscandia injection. A main point of our paper is that the two order magnitude longer typical paleoclimate simulations the reviewer is referring to are also difficult to interpret given their own sources of model uncertainty such as limited resolution, uncertainties and errors in boundary conditions/forcings, etc.
* * *
*The updates with former work is quite far from substantial, and the models used is very close to the one used in e.g. Condron & Winsor (2012). After almost a decade, computing power have strongly increased, so that these simulations now cannot be really considered state-of-the-art anymore, since far higher-resolution ocean-only simulations now exist, and do show that having even stronger resolution play a crucial role for the mean state of the AMOC (cf. Hirschi et al. 2020). As such, I am surprised that the authors still consider such short simulations (cf. point 1).*

We would argue that this study represents a significant improvement on previous work. As described in the introduction, there are three common issues in the design of experiments that implement freshwater in paleo contexts: 1) the freshwater is deposited directly over the sites of deepwater formation to compensate for inadequate horizontal resolution, 2) the amounts of freshwater used are unrealistically large, and 3) inconsistent/unrealistic ocean gateways. We also note that previous studies address aspects, but not all, of each of these issues. For example, Roche et. al., (2009) explored the impact of varying geographic regions, used appropriate gateways, and to a lesser extent used reasonable freshwater volumes but lacked the horizontal resolution to capture key transport features (e.g. boundary currents). Condron and Windsor (2012) and Hill and Condron (2014) addressed the horizontal resolution (3) and partially the geographic location issue

(2) but used unrealistic volumes (an order of magnitude larger) and had gateways and bathymetric features inconsistent with reconstructions (e.g. Barents-Kara being glaciated, assumption of eustatic sea level adjustment, among other limitations). They did not consider Fennoscandian and GoM freshwater sourcing nor did they consider the impact of open/close Barents Strait. We address 1) via releasing freshwater at coastal locations consistent with glacial reconstructions and by using a model well able to represent small scale features known to be important in the transport of coastally released freshwater. Issue 2) is addressed by bounding our fluxes by the upper limits of a self-consistent glacial reconstruction. Finally, 3) is addressed by using the relative sea level component of the self-consistent glacial reconstruction to configure our land-sea boundary and bathymetry, as well we address a limitation of the reconstruction by examining the impact a key gateway (the Bering Strait) has on our results for the most proximal injection location (the Mackenzie River). In summary, ours is the first such study to address these three common issues simultaneously and as such represents a significant improvement on previous work. These points are made clearer in the manuscript.

We point out the main focus of this work, the representation of surface transports and features generally regarded as subgrid scale, would not benefit from existing updates to the model as the features of interest are already adequately represented in the version we use. Updates to the model appear to largely center around bug-fixes and documentation updates (https://github.com/MITgcm/MITgcm/releases) without substantial effect on the representation of surface transports and eddies. As well, with regards to increasing the resolution of ocean-only simulations, we do note there are some entries in Hirschi et al. (2020) (which for the benefit of those unfamiliar with the work, is a review paper examining the representation of AMOC under present-day conditions from multiple sub 1 degree resolution model simulations extracted from 23 different publications) which are higher resolution. However, only one is a global ocean-only simulation which is above our grid resolution (Moat et al. (2016) which used 1/12 degree). Thus, we contend that the model configuration used in this study is of comparable complexity and resolution to the multi-model ensemble of simulations presented in Hirschi et al (2020). We make this point in the revised submission.

Furthermore, we would argue that the existence of higher-complexity or higher-resolution simulations for present-day phenomena does not negate the value of a study focussing on past oceanic phenomena using a model with slightly less complexity and lower resolution. The study here and those previous studies by Condron, Windsor, and Hill are still the highest-resolution, ocean-only simulations to date using bathymetry and boundary conditions that are not either pre-industrial or present-day (though the AWI group in Bremerhaven have conducted some very interesting paleo work with their unstructured high-resolution FESOM configuration). With regards to computing power having strongly increased, indeed some features of computing power have increased substantially but unfortunately model wall time does not decrease as per Moore's law as one might hope and enterprise computing focuses on parallel compute performance with a focus on stability, not single-thread performance, which does not translate into performance gains nearly as effectively.
* * *
*Line 90: such ocean-only model are simulations are not that costly within present-day computing time standard (e.g. Penduff et al. 2018 who considered 50 members of multi-decadal high resolution simulations…). Improvements as compared to former work with Condron as co-author, dating than almost a decade is not clear at all, while the main message remains also quite similar with this former work.*

It appears to us that the reviewer is making two separate claims in these comments. Firstly, they would like us to have run a larger ensemble of longer simulations or use a higher spatial resolution in the ensemble we did produce on the basis that such has been done in a previous study examining a different scientific question altogether. Secondly, they argue that the updates with former work (in the reviewer's words, "the use of glacial boundary conditions, and a more systematic analysis of the different potential outlet locations as well as the consideration of smaller rate of freshwater release, more in line with recent reconstructions.") are not substantial.

In regards to the first claim, we would argue that setting the bar for minimal requirements in an experiment to equal the most resource-intensive project published to date is illogical as doing so would rule out almost all researchers except those with the greatest access to resources. The simulations we have conducted represent an advancement over previous studies and are more than sufficient to provide important insight into the surface transport of continental runoff given we explicitly address 3 significant weaknesses in previous studies. Expecting us to greatly expand our simulation numbers and durations just because other multi-institutional projects have done so in other unrelated contexts is not reasonable nor necessary. As it is, these simulations occupied a substantial component of our computational allocation budget for the years during which they were run, the cost of which was O(10,000-100,000+CAD/year). Conducting over 100 years of simulation has consumed sizable compute resources unavailable to many researchers and required a Compute-Canada Resource Allocation Competition grant on the Niagara national system for both the storage (several hundred TB of data) and the compute time. Furthermore, we remind the reviewer how resolution, time-stepping, and compute-cost of a numerical model scales with resolution (generally cubic or higher (if the number of vertical levels is increased), such that a halving of horizontal resolution requires about 8x or more flops), Penduff (2018) having used a model roughly 33% coarser would be be able to execute their goals with more moderate compute resources (potentially even more so given no grid topology was provided for their experiments whereas we used a cubed-sphere topology which results in a generally uniform horizontal grid spacing of ~18km globally). Our experiments, when taken as a whole, are comparable to those presented in Hirschi et al. (2020).

As to the second claim, given we have conducted a study which explicitly addresses the primary drawbacks of the previous relevant works through "a more systematic analysis of the different potential outlet locations as well as the consideration of smaller rate of freshwater release, more in line with recent reconstructions." it would seem the reviewer contradicts their own claim of insufficiency. Three key limitations of previous studies, as stated above, severely limited what conclusions could be drawn from them. We have addressed those limitations.
* * *
*Improvements in our understanding of the impact of ocean resolution from models of oceanic circulation need to be more appropriately discussed (cf. Hirschi et al. 2020, Le Corre et al. 2020)*

A valid point, we included additional text to address this in the experimental design section (lines 108-115).
* * *
*The use of glacial boundary conditions apparently lead to a collapse of the AMOC in the ocean-only GCM used. The authors qualified it as a glacial state, but Fig. 6 shows a weakening AMOC index in the control simulation (which is thus not equilibrated at all) towards values of 2-4 Sv that rather correspond to an off state than a weak glacial states, according to e.g. Ganapolski and Rahmtorf (2001). AOGCMs indeed do not produce such weak state in glacial condition (e.g Kageyama et al. 2013, with all AOGCMs showing value larger than 5 Sv in their mean state). Considering an off state has major implications in term of barotropic circulation, notably in the subpolar gyre, which makes the relevance of those results doubtful for examining freshwater pathways at the beginning of e.g. the Younger Dryas as it is suggested in the paper.*
*Line 145-146: This claim is not supported by anu figures, and I strongly doubt of this, given the very small value at 26°N. The AMOC is state rather resemble an off-state. Can we see the meridional streamfunction in the last 10 years of the control simulation?*
*and*
*Line 152: "glacial mode" sounds very optimistic. The authors might need to discuss more what is known from data and models concerning the mean state of the AMOC during the LGM…*
*and*

*Line 153: "reasonable". This might be a bit too much optimistic as well I think. Please discuss appropriately the state of your AMOC, or provide more evidences to support that it can be considered as a glacial state.*
and
*The authors qualified it as a glacial state, but Fig. **S**6 shows a weakening AMOC index in the control simulation (which is thus not equilibrated at all)*

AMOC is not the focus of this paper and is only provided for context as it was expected portions of the community might seek information only on this metric despite the simulations here not being of long enough duration to make robust conclusions with regards to its behaviour (as further emphasised by the fact it was among the supplemental figures and not a central figure of the study as is common). Given this comment and others made, we clearly need to emphasise this further in the text. We emphasise this by the relocation of the AMOC discussion to the supplemental section S3 where it will not distract readers but remains readily accessible for those who are interested in the results despite the limitations of this metric for this study.

Furthermore the unclear language "Off-state" can be interpreted in multiple ways. If the reviewer is referencing the Off/Heinrich mode of AMOC operation as well summarised in Rahmstorf (2002) then we argue this is incorrect, as this mode of operation precludes the formation of deep water in the North Atlantic whereas Figure S5 clearly shows a robust mixed layer (note: this figure shows the average calculated over a full year, this reduces the magnitude of the mixed layer depth by comparison to a shorter averaging period like monthly maximum) . Furthermore, we note that 26N was chosen to correspond with the present day RAPID array and to allow for easier comparison to the previous work of Condron and Windsor (2012). This is not the location of the peak value of overturning in the North Atlantic basin as is typically reported in most investigations for whom AMOC is a constructive climate metric and thus the reviewer's comparisons to previous works based upon this value are unfortunately not readily accomplishable. Additional information regarding this value is now provided in the supplement, as there is a roughly -6Sv offset resulting from using 26N rather than the peak (that is, our AMOC maximum averages are ~9.5-10Sv rather than ~3.5-4Sv).

With regards to the effect that a reduced AMOC has on features closer to or at the surface, we find that the surface circulation tends to lead the deeper ocean in studies examining this coupling, not vice versa. One of the potentially most important surface features would be the subpolar gyre, which on glacial timescales modulates the salt transport to deep water formation regions (Klockmann, 2020). Furthermore, this coupling is found to be weaker in higher resolution eddy-permitting models than in coarser resolution models (Meccia, et. al, 2021), further reducing the impact of this feature. The other main surface feature of note which can be strongly affected by a weaker AMOC would be the Gulf Stream (as the reviewer has pointed out in another comment). Caesar, et. al., (2018) indicates that the latitude of the separation point of the Gulf Stream from the coast of North America is modulated by the AMOC, with a weaker AMOC resulting in the Gulf Stream shifting northwards and closer to shore. As raised in the other comment, this is now mentioned in the modified text. However, this does not change the impact of the Gulf Stream on our simulations or conclusions, whereby the Gulf Stream acts as an effective barrier to meridional transport of freshwater.

With respect to the structure of the AMOC, there was not a figure included as AMOC and its structure is not the focus of the paper nor is it relevant for inclusion in the manuscript given our primary interest is surface transport over short durations. Regarding the disequilibrium of the AMOC, it is indeed trending downwards initially but is relatively flat within the annual variability (one standard deviation is 1-1.5Sv as noted in the caption of Figure S6) for the years 10+ in Figure S6. Furthermore, we make no claims on the equilibrium of the deeper ocean whose equilibrium time is well understood to be several thousand years. The focus of this investigation is the very uppermost layers of the ocean (the majority of the anomaly is contained only within the top 30m of the ocean) whose equilibrium time is within the range of our investigation's duration.

Finally, to make clear that we do not consider analyses of the AMOC appropriate on the basis of these simulations, we added a corresponding line to the AMOC discussion section in the supplemental materials (Section S3).
* * *
*The discussion of the implications of their results for paleoclimate understanding is very weak and deserve to be strengthen. What does those results mean in regard to existing literature that GOM and GSL affect so weakly the convection zones? What does that mean in terms of last deglaciation storylines?*

*line 267: A proper discussion of the implications in terms of the storyline of AMOC changes over the deglaciation and the link with freshwater release should be provided. As an example, we can assume that those experiments strongly support a major role for freshwater release from Fennoscandia, as suggested in e.g. Toucanne et al. 2009. Please, further elaborate on this topic in light of existing literature.*

We have added additional discussion in the paper (lines 315-324) to address these points from the viewpoint of freshening of deep water formation regions rather than AMOC given the previously discussed de-emphasis of AMOC in our work.
* * *
*Line 99-100: more should be said concerning the experimental design. Since these are ocean-only simulations, how are considered the boundary conditions? Is there any SSS restoring? How evaporation fluxes are computed,*

We are not entirely clear what "more" the reviewer would like described in the experimental design section. However, between addressing the specific questions posed here (with relevant additions to the revised text, see Table S1 and lines 115-135) and those in Reviewer 2's review, we hope that we have satisfied the reviewer's request. There is no surface restoration, this would defeat the purpose of the experiments conducted. As per line 127 of the revised manuscript "We do not use surface restoration in our experiments. Evaporation is handled internally by the model in the EXF (external forcing package) from provided precipitation, relative humidity, and surface runoff fields."
* * *
*Line 134: this very zonal Gulf Stream might also be related with the fact that the AMOC is in an off-state, since this can strongly impact Gulf Stream pathway (e.g. Caesar et al. 2018)*

As discussed in the manuscript the zonal Gulf Stream is an artefact of the surface forcing, see plots below demonstrating the magnitude of the zonal component of our surface winds relative to a pre-industrial control simulation from CCSM4 (the closest model to what generated our original surface forcing). However a brief discussion of the northward/southward shifting of the separation point of the Gulf Stream from the East Coast of North America as a function of AMOC is now included with appropriate caveats in the supplemental section S3. Furthermore, we have conducted a sensitivity experiment replacing our glacial winds with that from the ERA40 reconstruction and find that this results in a less zonal Gulf Stream as expected. This additional information is available in the modified supplementary materials (Section S5).

[Figure]

Figure 1: The data are difficult to see during YD due to very strong red. Please consider another colour to allow proper examination of the curves

Accepted, color changed to black and line weight increased for better contrast.

Line 41 and elsewhere: "eg." Should be replaced by "e.g."

Accepted

Line 89-90: How many vertical levels in the model?

The model features 50 vertical levels, this is now noted in the model description section (line 100).

Line 141: "Labrador" Sea (not sea)

Fixed.

Line 225: "yr" is not defined.

yr is the CP style guide requested abbreviation for year. Defining this is not requested by the style guide and can be understood from context.

Line 267: An additional caveat is not properly discussed which is the fact that the authors consider here ocean-only model, which prevent from considering any potential coupled ocean-atmosphere feedback, which might play a role.

A useful suggestion, a brief discussion of this point has been added (lines 180-185)

*Line 268: "under Younger Dryas conditions": this statement does not really reflect the off state that is simulated in the control simulation.*

A weakened AMOC (McManus, et. al, 2004) and stadial surface conditions are very much the expected configuration of a Younger Dryas climate (Carlson A.E., 2013).
* * *
*Line 276-279: it is quite unclear from where those estimates come from, which is weird to provide in the conclusion, since not shown in the result section. I assume, they are estimated from a similar approach as in line 232-241, which is considering an ocean without any circulation at all. This is quite a strong hypothesis… Thus, I'm not sure those estimates are really useful, especially in the conclusion.*

They are indeed estimates using the same simple method as in lines 232-241 (pre-print version), this has been made clearer in the conclusions. We chose the simpler of assumptions when making these 'back of the envelope' estimates, as the alternative would be to assume some structure of flow under a regime for which we do not have data (one would expect 2dSv of freshwater into a region to affect transport in/out of a region) and could easily scale the results for dramatic impact by making such assumptions. As noted on lines 237-238 (pre-print version) this approach is reasonable for a simple estimate, as we find our calculated salinity anomaly for Fennoscandia to be comparable to the simplified hosing estimate within the first year. This localized hosing which we compare to is something which has been done before in previous investigations (see Roche, et. al., 2010)  and hence why it was done for comparison.
* * *
*Line 283: "better ways to mitigate this problem": this sentence is quite enigmatic. Can you please clarify what is meant here?*

This is the subject of upcoming work which is outside the scope of this manuscript. We have modified the text to express that this is the subject of upcoming work.
* * *
*Line 284-285: it should be stated here that these investigations are done in an off-state for the AMOC, and during only 20 years.*

AMOC is not the focus of this paper and is only provided for context (see previous discussion) and is not a relevant discussion point for the conclusions of this paper. The duration of the investigations are already described in the body of the paper and associated figures. We make clear that the AMOC state is reduced glacial model (it is not off).
* * *
**References:**

Carlson A.E. (2013) The Younger Dryas Climate Event. In: Elias S.A. (ed.) The Encyclopedia of Quaternary Science, vol. 3, pp. 126-134. Amsterdam: Elsevier.

Klockmann, M., Mikolajewicz, U., Kleppin, H., & Marotzke, J. (2020). Coupling of the subpolar gyre and the overturning circulation during abrupt glacial climate transitions. Geophysical Research Letters, 47, e2020GL090361. https://doi.org/10.1029/2020GL090361

McManus, J. F., Francois, R., Gherardi, J. M., Keigwin, L. D., & Brown-Leger, S. (2004). Collapse and rapid resumption of Atlantic meridional circulation linked to deglacial climate changes. nature, 428(6985), 834-837.

Meccia, V.L., Iovino, D. & Bellucci, A. North Atlantic gyre circulation in PRIMAVERA models. Clim Dyn (2021). https://doi.org/10.1007/s00382-021-05686-z

Rahmstorf, S. (2002). Ocean circulation and climate during the past 120,000 years. Nature, 419(6903), 207-214.

Roche, D. M., Wiersma, A. P., & Renssen, H. (2010). A systematic study of the impact of freshwater pulses with respect to different geographical locations. *Climate Dynamics*, *34*(7-8), 997-1013.

**RC2 Point-By-Point Reply:**

Firstly we would like to thank the anonymous reviewer for the time they have taken to read through the manuscript and make their suggestions. An itemized set of responses to their comments is below, beginning with the more substantial points and technical points following.

*-The main story and motivation of the study is hidden behind all the text. It should be re-written with a clear and standard structure, where general description of the problem, goal of the paper and previous studies are clearly discussed.*

As submitted for discussion, the paper introduction does follow a conventional structure (context, motivation, previous work and its shortcomings, and goals of the present manuscript). We also believe the abstract succinctly summarizes the main story. However, it seems this structure isn't coming out clearly for the reviewer. For this reason, we added some additional guiding sentences and words to make the logical argument clearer to all readers.  Examples of the changes we made include:

'The goal of this study is to directly address all of these limitations...'
'We start our discussion of the experimental design with a brief overview of the model configuration…'
'Here we discuss the heredity of our simulations'
* * *
*-Figure 1, is a very nice figure but it is not really discussed and not clear why you used it. I think it deserves more explanation.*

We have added additional contextual information to better make use of Fig. 1 (see lines 315-324)  as well as expand the introduction to address the first issue the reviewer raises. As well, to increase the utility of Fig. 1 we have merged in the runoff flux figure from the supplement and placed it on the same time scale so as to better discuss our results in the context of the last deglaciation.
* * *
*-The model description should be improved (e.g., you need to clearly specify that you use a global model and how many vertical levels your model has). Next, discuss the forcing. Then, explain the initialization of experiments, the control runs, number of spin up years (exact numbers), and total simulation years. Last, explain the experiments with all the needed details. In the current version, you might have given most of these information but it is done in a messy way.*

We note the global nature of the model grid as well as the vertical level count in our revisions, as well we have made the heredity and durations of each of the experiments clearer via a table in the supplemental material.

We agree with the reviewer that the Experimental Design section could be structured better.  However, we don't find the structure the reviewer suggests to be very helpful.  Instead, we add additional information (lines 116-128) to the model description as requested, and then discuss the control simulations (initialization and forcing), followed by the freshwater injection runs (initialization and forcing).  Since not all of the simulations are of the same duration, these types of details bog the text down.  Instead, we've added a table to the supplement (Table S1) that specifies such details for each run. Furthermore, as with the introduction we have added additional guiding sentences.
* * *
*-I think your Figure S2 should be discussed in this section and be used as a main figure and not a supplementary.*

We have merged the useful information of Fig. S2 with Fig. 2.
* * *
*- I wonder why there is no summer sea ice in the Arctic in the region above (north of) Greenland? As far as I know that is a region that is covered by sea ice in summer (for present day condition).*

We have noted this in our discussions as well. To the best of our knowledge, it is not an artefact of the surface forcing, which was the most likely candidate, but rather an artefact of the time-domain averaging chosen. We previously used time max/min, and so values of zero sea ice at any given point in the last 5 years would result in what appears to be a cell with zero sea ice. We have addressed this via using monthly means averaged over the last 5 years with February corresponding to the maximal extent and August corresponding to the minimal extent. A slightly lessened sea ice concentration is also coincident with increased vertical mixing in the area (this can be seen in the mixed layer depth contour of Figure 2) but it is unclear if this is a result or the cause of this feature. However, digging further into this question lies outside the scope of this paper. Finally, we note that our sea ice extents are largely consistent with reconstructions (e.g. de Vernal, et. al., 2005)

A. de Vernal (2005) Reconstruction of sea-surface conditions at middle to high latitudes of the Northern Hemisphere during the Last Glacial Maximum (LGM) based on dinoflagellate cyst assemblages, Quaternary Science Reviews, https://doi.org/10.1016/j.quascirev.2004.06.014.
* * *
*- Figure S5: you show only 1 year (the last year of simulation). Please choose similar intervals and same number of simulation years for all the figures (e.g., 5-y mean of year x to y).*

Easily done. All figures are shifted to reflect the last 5 simulation years.
* * *
*Same for Figure 2, what is meant by single day? For your study, yearly-mean values should be fine but take the average over several simulation years.*

We used quite literally a single day (daily mean to be more precise, we have added this language to enhance clarity in the manuscript) to more readily demonstrate the turbulent nature of the model. We had multiple versions of that figure using various other temporal averaging schema but none conveyed the point as well as a single day. Eddies and variations in the coastal boundary currents are readily 'averaged away' when considering any sort of time-averaging.
* * *
*- Would be interesting to see the timeseries plot for the MLD in Labrador and Nordic seas.*

We had generated similar plots but they did not convey useful information at the time (anomaly does not show a trend), upon revisiting this idea with a smaller domain as per the reviewer comment this conclusion is unchanged.
* * *
*- Figure S6-AMOC: You initialize the model from the experiment by Hill and Condron (2014), right? But why your experiment's AMOC is about 6 Sv at year -10 while it should be larger given the AMOC in Hill and Condron (2014)? If I am mistaken, please explain this part better.*

Not quite, our run was initialized from a re-run of the Hill and Condron control simulation (very early versions of this work required additional data not available from the Hill and Condron simulations so a re-run was required). After this rerun, we used the temperature and salinity fields from the 20th year of the LGM simulation to initialize the Younger-Dryas-like configuration (a direct restart was not possible due the significant bathymetry changes). The Younger Dryas configuration was run forward for 10 additional years, after which we then branched the two control runs. We make this information clearer in the revised version to address this comment as well the previous comment regarding experimental design (see lines 116-128 and Table S1).

*Overall, the AMOC in your study seems to be smaller than some similar studies (https://agupubs.onlinelibrary.wiley.com/doi/pdf/10.1002/2015GL064583), right?*

Between what other comparable studies we have found (e.g. Hirschi et al. 2020) and what the reviewer has referenced we do find our AMOC is among the weaker values we have found in the literature. We do note that our use of 26N to correspond with RAPID and Condron and Windsor (2012) makes our model seem weaker relative to some other studies as that is not the peak of the stream function in the Atlantic as is commonly used for studies where AMOC is a useful climate metric. When we use the peak of the stream function in the Atlantic we find our values to be ~9.5-10Sv for the last 5 years of the CBS control vs. ~3.5Sv at 26N.

*What is the difference in AMOC between CBS and OBS control runs? The AMOC difference between the experiments seems to be small (1 or 2 Sv), and perhaps within the range of model internal variability. I am not sure if the AMOC can give any conclusive view.*

As noted, the difference in values is very small relative to the range of model variability. When using the peak of the stream function in the Atlantic averaged over the last 5 simulation years we find that CBS control run has an average of 9.5Sv vs the OBS control run with 8.6Sv. We agree that the AMOC can not give a conclusive view of what is going on in the simulations. Due to this we have presented the AMOC only for context as we anticipated leaving out discussion of AMOC in this manuscript would raise calls for its inclusion. We moved all discussion of AMOC in our simulations to supplementary materials (section S3) to better emphasise this idea and to address concerns raised by the other reviewer.
* * *
*-Figure 4: Is it surface salinity? Except in the middle panel, the salinity anomaly shows a downward trend in some of the experiments (for instance the CBS MAK). You need to be careful how you interpret these as the model is not clearly far from equilibrium.*

This is indeed sea surface salinity, more specifically the salinity from the top layer of our model (10m thickness). Given the trends, we elected to integrate our simulations further forward within the limitations of our resources (for most runs this amounts to ~5-10 additional simulation years). This additional model integration does not affect most of our conclusions, as the relative ordering of freshening at the different regions investigated remains the same with the exception of a larger freshening effect from the OBS MAK simulation for the NADW region. This additional information is reflected in the updated manuscript (primarily Figs. 3 and 4).
* * *
*-One implication of this study is for the Younger Dryas event which is linked to temperature changes, and I was expecting to see a plot for the sea surface temperature (SST). I realized that this an only ocean model study but would still be interesting to see the (indirect) impact of different FW injection on SST.*

We include a SST anomaly plot for the 2dSv CBS Mackenzie River simulation using the same time period (last 5 simulation years) below for the reviewer and others interested (this figure is also duplicated in the updated supplemental material). Interestingly the distribution generally follows the salinity anomaly distribution but the extreme values of negative salinity results in warming relative to the control while more saline values results in a cooling, with the threshold between warming and cooling being ~-2PSU. We do not have an immediate use for this data in this manuscript outside providing some small context and so invite any interested parties to contact the authors at a later date if seeking further discussion.

**SST Anomaly - Last 5 simulation Year Average**
**2dSv Mackenzie River - CBS**

[Figure]

**Sea Surface Temperature Anomaly (K)**
* * *
*-I will include a figure similar to Figure 3 but for BS closed in the main paper.*

We only have the one closed Bering Strait freshwater forcing simulation and so cannot create something akin to Fig. 3 but with Closed Bering Strait runs. Given the broad scale similarities between the OBS and CBS 2dSv Mackenzie River injection scenarios we will leave the CBS run in the supplement only.
* * *
*Also Figure S6 is better to be in the main paper.*

We reiterate the point that the focus of our work is the transport of freshwater at the surface of the ocean. Including a figure on AMOC (which in our study is not a reliable metric given the short timescales of spinup) would reduce clarity in the manuscript and needlessly focus the reader on one of the more unreliable aspects of the study. As noted by the reviewer and ourselves, the AMOC in our simulations can not give conclusive results. As such we will keep Fig. S6 in the supplementary materials.
* * *
*Title: needs to be adjusted. Is it really during the last deglacial?*

The freshwater forcing fluxes are bounded by those from the Younger Dryas period of the Tarasov GLAC reconstruction, as well the bathymetry is derived from the same. Given the only glacial element of our

configuration is the surface forcing we consider deglacial to be the more accurate description. However, the title has been revised given this comment and others made in the discussion phase.
* * *
*Line 9: You are using paleo forcing and paleo-bathymetry, please correct it.*

The line in question is : "We focus particularly on the **prior use of excessive freshwater volumes** (often by a factor of 5) **and present-day** (rather than paleo) ocean gateways…" (emphasis ours)
The statement shows that we contrast our work relative to previous investigations who used present-day bathymetry.
* * *
*Line 35: Sv is the common unit to use, and dSv is not really helping to make things easier.*

Indeed Sverdrup is a common unit when discussing the AMOC and hosing. However, 1Sv is an order of magnitude (or more) larger than is reasonable for freshwater outflow from glacial runoff, whereas dSv (1/10 Sv) is the same order of magnitude of fluxes presented in this work as well as upcoming work by the authors. Previous works could reasonably use Sv as many used values of freshwater flow that were O(1Sv) but given reconstructions preclude these values we advocate for using something more easily read (decimal points are quite easy to miss). For clarity and continuity we will continue to use the more appropriately scaled unit, but we have added a line in the text when we define dSv explaining why we prefer this less common formulation (line 41).
* * *
*Line 91-92: Does the model really captures the coastal boundary currents?*

Both Yang (2003) (10.1016/S1463-5003(02)00058-6) and Nurser and Bacon (2014) (10.5194/os-10-967-2014) provide ranges of boundary current widths well within the range of our model's horizontal resolution. As noted in the paper this becomes problematic at the highest latitudes due to the significant decrease in the Rossby radius at high latitudes. Despite this we do see the freshwater constrained to very narrow boundary currents even in the Arctic.
* * *
*Line 104: "The first…": revise*

We have clarified this text.
* * *
*Line 134: "...discussed in Experimental design section": is not discussed*

Surface forcing is listed on Line 98 (pre-print version) in the Experimental Design section:
"The surface forcing used in that simulation includes winds, precipitation, 2m atmospheric temperatures, short and longwave radiation, surface runoff, and humidity from the CCSM3 working group's contribution to PMIP2 (Braconnot et al., 2007)."
* * *
*Figure S6: It is strange to use negative time values for the model spin up period.*

Given all our runs are relative to each other, the use of negative for a period of time which is only discussed in one figure allows for comparison to Fig. 4 without that figure being required to start at year 10.

**CC1 Point-By-Point Reply:**

Firstly we would like to thank the community reviewers/commenters for freely offering their time to read through the manuscript and make their suggestions. An itemized set of responses to their comments is below, beginning with the more substantial points and technical points following.
* * *
*The major point of the paper is simple enough. Freshwater released in realistic locations, with realistic circulation and at realistic rates doesn't reduce deep water formation.*

It was very helpful to us to see what the reviewers consider the major point of our paper, since it's not what we intended.  We would describe the major point of the paper as, "The salinity anomalies introduced over deepwater formation regions (and thereby potential for deepwater response) by freshwater released from realistic locations at realistic amounts differ depending on the location."  Having become aware of this confusion and similar comments by the other reviewers, we've made sure this comes out more clearly in the revised manuscript.
* * *
*With respect to the title, it reads as methodology, not as a main finding.*

The title is a WYSIWYG (what you see is what you get) title which explicitly conveys the subject of study. An internal discussion has been had with regards to the title to see if we can convey the same information while broadening the audience and appeal, the result of this discussion to we have chosen to change the title to "Northern Hemisphere freshwater routing in eddy-permitting simulations of the last deglacial"
* * *
*With respect to the abstract, we feel that it is much too long.*

 We disagree and find that shortening the abstract any further would result in the loss of useful information.
* * *
*There are many acronyms in this paper. As it is a short paper, we think these should be spelled out more to ease the readability.*

The authors note the only acronyms used in this paper that are not well established in the community are those used for the model runs or features of the model runs (e.g. MAK, GSL, GOM, FEN, CBS, OBS), and those for subjects more readily identifiable by their acronyms by those in the field (e.g. ICE-5G, GSM, MITGCM, PMIP, CMIP). Spelling out each acronym for each occurance would lead to unnecessary bloat of the paper and would thus decrease clarity.
* * *
• *In the introduction, it is mentioned that there are "at least three common experimental design problems", but the authors only expand on two.*

We have clarified the text to reflect the three common experimental design problems better.
* * *
• *It would be nice to have some information in the introduction for why the authors chose the sites they chose. Are these areas known to be the major outflows of freshwater during the glacial? Are there others that are not accounted for?*

The Mackenzie river, the Gulf of St. Lawrence, and the Mississippi/Gulf of Mexico are well established outlets for glacial runoff during the last glacial cycle, these were the most prevalent for liquid flux off the North American ice sheets (this has been clarified with a modified figure 1, including information from supplemental

figure 1). The other main outlet, which we do not explore in the manuscript (but is discussed) but was addressed in Condron and Hill (2014) is Baffin Bay, which is primarily solid flux (i.e. Icebergs). This additional information has been emphasized in the introduction as motivation for the sites chosen (lines 38-40). We invite the community reviewer to explore

Tarasov, Lev, and W. R. Peltier. "A calibrated deglacial drainage chronology for the North American continent: evidence of an Arctic trigger for the Younger Dryas." Quaternary Science Reviews 25.7-8 (2006): 659-688.
* * *
 • *It would also be nice for the introduction to talk more to why an AMOC collapse is thought to have occurred many times in the past. It is implied, but not clearly stated in the introduction. It is also not stated why we might be interested in AMOC collapse today, which may be obvious to the authors but would be worth stating.*

Given that the above would be standard knowledge for anyone in the paleoclimate field, such an addition would detract for most readers. However, to facilitate access for other readers, we have added a few more references addressing the above.
* * *
• *Another topic that is not mentioned is the bistability of the AMOC. There has been much work on the existence of "tipping-points" (e.g. most recently Lohmann & Ditlevsen, 2021, PNAS), whereby over a certain threshold of freshwater hosing the AMOC collapses, but underneath that threshold it does not. This is an important concept to include given that despite some significant freshening in your experiments there is little effect on the AMOC.*

We note that the simulations are not of sufficient duration to obtain a robust AMOC signal, and AMOC was provided only for context and comparison to Condron (2012). As such, discussion of bistability and AMOC are best left to other studies, one of which is upcoming, whose timespans can more readily and reliably distinguish these features.
* * *
• *The ice extents in km2 are quite low given that the record minimum in 2017 was 14.3 million km2 .*

We point the community reviewer to the land-sea mask in Fig. 3, where all of the Canadian Arctic Archipelago is glaciated and sea level has also been reduced, thus reducing the area over which sea ice can form in the model relative to present day conditions.
* * *
• *Line 170 should read ". . down to 200 m depth. . ." [This comes of too much use by scientists of "high" instead of "large"]*

Addressed.
* * *
• *Line 186. Direct transport from FEN across the GIN seas is not clear in the figure, the proportion of fresh water shown is very small.*

Unsure how to address that a still figure cannot readily convey motion as was observed by the authors from examining the model output. Hence why this motion was described in the manuscript. Indeed the proportion of freshwater that follows this path is quite small.
* * *
 • *Line 191 becomes clear looking at the figure but there has to be meridional transport to get from the Gulf of St. Lawrence to the Gulf Stream and it is curious to read "eastern . . North Atlantic"*

Comment noted, no action required.

• *Lines 229-231. A curious statement. As though the  meltwater is trying not to affect AMOC.*

Comment noted, no action required.
* * *
• *It should also be noted that the authors did not complete a combined experiment where all sites received increased freshwater fluxes at the same time. This might have been sufficient to tip the AMOC into a collapsed state. At the very least, this should be discussed. At most, another simulation should be performed with all four release sites simultaneously releasing freshwater.*

This paper is not a direct investigation of AMOC, the timescales involved in our simulations are neither long enough to sufficiently equilibrate the deeper ocean layers nor to observe a robust AMOC response. The use as a study for AMOC (and other climate effects) instead is the subject of upcoming worth from the authors using a lower resolution model and longer simulations. We constrain our present freshwater injections to that within the realms of reality, a sustained 8dSv into the ocean at a singular outlet is not a realistic flux under any robust deglacial reconstruction if this is what the reviewer is referencing. If the reviewer is instead suggesting a reduced flux (such that it is bounded by a reconstruction like was used in the manuscript) but from all locations at once, this would indeed be interesting but more useful to explore the (non)linear effects of runoff from the outlets. While interesting this is more appropriately explored as a separate investigation where all combinations can be thoroughly examined. However, a related question raised by this comment is whether there is any reason to expect that for the freshwater flux indicated, are the salinity anomalies purely passive tracers or do they feedback on the near surface wind-driven circulation for the order decadal time-scale (ie much shorter than AMOC timescale)?  If they are purely passive, then one would expect the routing response to be linear. Though the injection does slightly change near surface density and sea surface height, this is miniscule compared to monthly and yearly variations in windstress (perhaps not the case for the original Condron/Winsor experiment with factor 25 larger freshwater fluxes). Though still unlikely to be significant, a non-linear active response assessment would require fully coupled atmosphere-ocean modelling to properly assess, and therefore beyond the bounds of this study.
* * *
• *We suggest a change in the colour scheme of figures from jet to something more colour-blind friendly.*

We have adjusted all figures using GMT's WYSIWYG scheme (a rainbow-esq colour scheme) to the same color-blind-friendly scheme used in Figure 2.
* * *
• *Figure 1 is confusing and needs more details. We do not know how to interpret it.*

Given this figure has been highlighted as underutilized by another reviewer and the community reviewer requests more details we have added additional text to make use of this figure. As well we have provided some guiding text (in Figure 1's caption) for those readers less familiar with paleoclimate proxies.

For additional resources and background on the concepts of  interpreting δ18O time series/temperature reconstructions from ice cores, and relative sea level , we suggest reviewing:
Shennan, I., Long, A. J., & Horton, B. P. (Eds.). (2015). Handbook of sea-level research. John Wiley & Sons.
For foundational information on relative sea level and
Cronin, T. M. (2009). Paleoclimates: understanding climate change past and present. Columbia University Press.
For all other paleoclimate resources (particularly for δ18O, its uses and limitations as a paleoclimate temperature proxy) as well as the latest IPCC assessment report. All of these have proven very helpful for

developing a foundation in these subjects and ought to be very useful for digesting other subject material which make ready use of paleoclimate proxies.

---

## Referee Report (RR1)

**2nd Review: Freshwater Routing In Eddy-permitting Simulations Of The Last Deglacial:Impact Of Realistic Freshwater Discharge, by Love et al.**

Compared to the earlier version, the revised manuscript has been improved but I still have some rather minor comments which are listed below. Please not that, I used the file with track changes and the line numbers refer to that file.

Title: Needs modification. You do not simulate the Last Deglacial

Line 9-11: "We focus particularly…" Not clear, re-write. Also better to merge this sentence with lines 13-15.

Line 41-42: "The more than …" sentence is incomplete or missing something

Line 95: "...has been shown to have global impacts…" replace global with large-scale. You can also cite Karami et al. (2021) where they study the impact of Arctic gateways.

Note that you often use two sets of brackets to cite a paper instead of one set.

Line 196: I do not know if it helps your arguments here, but in Karami et al. (2021) they show that gateway fluxes, surface properties and the sea ice reach semi-equilibrium after 35 years (they use a different model though).

Line 232: Please explain why the OBS simulation has a larger sea ice extent?

Lines 280-290: results concerning CBS versus OBS are consistent with Karami et al. (2021). Please see the mentioned reference for the explained dynamics as well as the results for closing/opening CAA.

Line 347: what do you mean by "with the possible exception"?

Line 409: "larger impacts" largest?

Any suggestions/comments about what we can learn from this study for the future climate?

Reference:
Karami, M.P., Myers, P.G., de Vernal, A. et al. The role of Arctic gateways on sea ice and circulation in the Arctic and North Atlantic Oceans: a sensitivity study with an ocean-sea-ice model. Clim Dyn (2021). https://doi.org/10.1007/s00382-021-05798-6

---

## Author Response (AR3)

This is my second review of this paper. Generally speaking, the authors have correctly accounted for my comments. The manuscript is now far clearer and focused.

Thus, I believe the paper is worth publication and I only have some minor comments.

• Line 20: "ocean circulation": as the authors explain it clearly, there are not focusing much on the ocean circulation, but more on the ocean freshening. I propose to replace by "oceanic freshening and potential impacts on the ocean circulation"
• Line 41: replace "e.g" by "e.g."
• Line 65: replace "ie." By "i.e."
• Line 95: might be worth to define the acronym "yr" here.
• Line 145: "similar structure". I cannot find any figure of the AMOC structure. Please add a "(not shown)". Also, since the intensity is about 5 times weaker, maybe you can add "far" before "weaker".
• Line 276-279: , although I already mentioned this in my former review, I still have the feeling that the numbers provided here are not well depicted, and it might be difficult for a reader to redo the calculation from the authors. This is an issue for a science paper, so please, develop the computation leading to those estimates somewhere in the paper, before the conclusions, or in a supplementary.
• Fig. S4: I think it might be necessary to mention that what is shown is sea concentration, with unit varying from 0 (no sea ice) to 1 (fullly sea-ice covered)
• Fig. S6: Please add the unit on the y-axis

**Referee #2 - Pasha Karami**

Compared to the earlier version, the revised manuscript has been improved but I still have some rather minor comments which are listed below. Please not that, I used the file with track changes and the line numbers refer to that file.

Title: Needs modification. You do not simulate the Last Deglacial
Line 9-11: "We focus particularly..." Not clear, re-write. Also better to merge this sentence with lines 13-15.
Line 41-42: "The more than ..." sentence is incomplete or missing something
Line 95: "...has been shown to have global impacts..." replace global with large-scale. You can also cite Karami et al. (2021) where they study the impact of Arctic gateways.
Note that you often use two sets of brackets to cite a paper instead of one set.
Line 196: I do not know if it helps your arguments here, but in Karami et al. (2021) they show that gateway fluxes, surface properties and the sea ice reach semi-equilibrium after 35 years (they use a different model though).
Line 232: Please explain why the OBS simulation has a larger sea ice extent?
Lines 280-290: results concerning CBS versus OBS are consistent with Karami et al. (2021). Please see the mentioned reference for the explained dynamics as well as the results for closing/opening CAA.
Line 347: what do you mean by "with the possible exception"?
Line 409: "larger impacts" largest?
Any suggestions/comments about what we can learn from this study for the future climate?
Reference:
Karami, M.P., Myers, P.G., de Vernal, A. et al. The role of Arctic gateways on sea ice and circulation in the Arctic and North Atlantic Oceans: a sensitivity study with an ocean-sea-ice model. Clim Dyn (2021). https://doi.org/10.1007/s00382-021-05798-6

**Response to Anonymous Referee #1**

Again we would like to thank the anonymous reviewer for the time they have taken to re-read the manuscript. Given the comments are either minor or technical, we will put responses in-line with the provided comments. We note that the line numbers provided by the reviewer do not seem to consistently align with the documents we provided to CP (presumably just an odd technical problem) and so we address the feedback as best as possible given that limitation.

• *Line 20: "ocean circulation": as the authors explain it clearly, there are not focusing much on the ocean circulation, but more on the ocean freshening. I propose to replace by "oceanic freshening and potential impacts on the ocean circulation"*
We do not find any references to 'ocean circulation' around line 20, and the suggested changes do not seem to fit the two instances in the document which use this language.

• *Line 41: replace "e.g" by "e.g."*
All found version are fixed

• *Line 65: replace "ie." By "i.e."*
All found version are fixed

• *Line 95: might be worth to define the acronym "yr" here.*
Replaced "yr" with "years". It is only used 4 times in the document so seems a decent compromise between following CP guidelines and enhancing readability.

• *Line 145: "similar structure". I cannot find any figure of the AMOC structure. Please add a "(not shown)". Also, since the intensity is about 5 times weaker, maybe you can add "far" before "weaker".*
Not found in the main text, but addressed where relevant in the supplementary material.

• *Line 276-279: , although I already mentioned this in my former review, I still have the feeling that the numbers provided here are not well depicted, and it might be difficult for a reader to redo the calculation from the authors. This is an issue for a science paper, so please, develop the computation leading to those estimates somewhere in the paper, before the conclusions, or in a supplementary.*
Added to the supplementary material and referenced in the main text.

• *Fig. S4: I think it might be necessary to mention that what is shown is sea concentration, with unit varying from 0 (no sea ice) to 1 (fullly sea-ice covered)*
Added the following line to the caption (axis title already displays concentration):
"Sea ice concentration values range from 0 (no sea ice) to 1 (full grid cell coverage)"

• *Fig. S6: Please add the unit on the y-axis*
Figure S6 is not a time series with a y-axis to which we can add units. Figure S11 seems to be missing a title and units (removed by auto-cropping), and this is now fixed.

**Response to Referee #2 - Pasha Karami**
We would like to thank Pasha Karami for taking the time to re-read the manuscript and suggest further improvements. Given the comments are either minor or technical we will put responses in-line with the provided comments.

*Title: Needs modification. You do not simulate the Last Deglacial*
The description of "Last Deglacial" is an accurate description of the model configuration we use considering the compromises in the model boundary conditions. In our view, this is the most reasonable characterization as neither adjacent time intervals, "Last Glacial Maximum" or "Holocene", are better representations of the simulation configuration.

*Line 9-11: "We focus particularly..." Not clear, re-write. Also better to merge this sentence with lines 13-15.*
We disagree with the comment on merging as that would be blending limitations of previous work with our results and would reduce clarity. As for being unclear, the sentence in question is as follows:
"We focus particularly on the impact of: 1) the injection of freshwater directly over sites of deep-water formation (DWF) rather than at runoff locations, 2) excessive freshwater injection volumes (often by a factor of 5), and 3) the use of present-day (rather than paleo) ocean gateways."

This characterizes the common limitations of previous studies which we address in this study with additional details provided in the introduction. We have added 'Hosing' to point 1 to try and assist clarity but the other points are very specific and add sufficient contextual information with regards to each limitation (within the limited scope of an abstract).

*Line 41-42: "The more than ..." sentence is incomplete or missing something*
The sentence in question now reads as follows,
"The onset of the subsequent cold Younger Dryas interval occurs more than a millennium later, which is longer than would be consistent with a direct physical linkage."
We hope this addresses the concern the reviewer has with this sentence.

*Line 95: "...has been shown to have global impacts..." replace global with large-scale. You can also cite Karami et al. (2021) where they study the impact of Arctic gateways.*
"Large-scale" is imprecise relative to "global" and would reduce clarity. Leaving text in question as "global".

*Note that you often use two sets of brackets to cite a paper instead of one set.*
Fixed

*Line 196: I do not know if it helps your arguments here, but in Karami et al. (2021) they show that gateway fluxes, surface properties and the sea ice reach semi-equilibrium after 35 years (they use a different model though).*
Interesting information, however we don't believe we can include that information in a way which would be of benefit to a reader due to the differences in the model configuration and initial conditions.

*Line 232: Please explain why the OBS simulation has a larger sea ice extent?*
This was not explicitly investigated in the study, however there are two differences we observed while examining other aspects which may account for this difference. Firstly, the OBS configuration has sea ice at the Bering Strait rather than land, which results in a bias towards the OBS configuration when comparing these numbers. Secondly, the other main region of difference with respect to SIA is east of the island of Newfoundland. The OBS configuration has stronger western boundary currents in the Arctic and North Atlantic. This ought to lead to greater sea ice export from the Arctic, which tends to transport the sea ice towards this

region. Both of these features are readily seen in Fig. S4. These have been summarised via an additional sentence after the quoted line.

"Sea ice extent is larger in the OBS simulation due to two features: the expanded ocean area surrounding the open Bering Strait, and enhanced sea ice export, see Fig. \ref{sfig:sicMerge}."

*Lines 280-290: results concerning CBS versus OBS are consistent with Karami et al. (2021). Please see the mentioned reference for the explained dynamics as well as the results for closing/opening CAA.*
The reference appears to be useful for further investigations of this type. We have added the following line.
"When comparing against OBS/CBS results from \citet{Karami_2021} (with the closed Canadian Arctic Archipelago), it is apparent that this contrast is present even in unforced simulations."

*Line 347: what do you mean by "with the possible exception"?*
Text modified for clarity, now reads:
"None of the simulations appear to have reached equilibrium in the North Atlantic with the exception of FEN. The prominent seasonal cycle of FEN, which exhibits the largest amount of variability on inter-annual timescales, reduces confidence in this conclusion."

*Line 409: "larger impacts" largest?*
Larger is most appropriate, largest would be OBS-MAK.

*Any suggestions/comments about what we can learn from this study for the future climate?*
Given the plethora of future climate studies, the significant boundary condition differences, and the large uncertainties associated with this study (relative to present day investigations with better constraints), we find extrapolating our results to future climate studies to be problematic. However, one finding of note which should be applicable for both intervals is the blocking/entrainment action of the Gulf Stream which reduces the meridional transport of freshwater. The tilt of the Gulf Stream is very different, but the overall action ought to be the same, barring a large weakening of the current. This is reflected via the additional information added to the conclusions of the manuscript:

"As well, the reduction of meridional transport of freshwater across the Gulf Stream observed in our results is a feature which ought to be equally applicable, and considered when not explicitly resolved, for both paleoclimate and future climate investigations."

**Additional Note:**
There was some re-evaluation by one of the co-authors of a dataset referenced in the manuscript (some of the Tarasov et. al., 2012 data), which resulted in reverting references to an earlier (and considered by the co-author, more robust) form (Tarasov & Peltier, 2006) and modifying Fig. 1. This does not affect any of the conclusions of the work but does involve some changes to the text to reflect this shift to the different dataset(s).

In the process of addressing the minor revisions we, the authors, noted some sentences that could be tweaked to increase clarity for the reader.

[revised manuscript text omitted]